# Highly-stable, injectable, conductive hydrogel for chronic neuromodulation

Ming Yang[1,4], Lufang Wang[2,4], Wenliang Liu[1,4], Wenlong Li[1], Yewei Huang[3], Qiaofeng Jin[2], Li Zhang[2] ✉, Yuanwen Jiang [3] ✉ & Zhiqiang Luo [1] ✉

Electroceuticals, through the selective modulation of peripheral nerves near target organs, are promising for treating refractory diseases. However, the small sizes and the delicate nature of these nerves present challenges in simplifying the fixation and stabilizing the electrical-coupling interface for neural electrodes. Herein, we construct a robust neural interface for fine peripheral nerves using an injectable bio-adhesive hydrogel bioelectronics. By incorporating a multifunctional molecular regulator during network formation, we optimize the injectability and conductivity of the hydrogel through fine-tuning reaction kinetics and multi-scale interactions within the conductive network. Meanwhile, the mechanical and electrical stability of the hydrogel is achieved without compromising its injectability. Minimal tissue damage along with low and stable impedance of the injectable neural interface enables chronic vagus neuromodulation for myocardial infarction therapy in the male rat model. Our highly-stable, injectable, conductive hydrogel bioelectronics are readily available to target challenging anatomical locations, paving the way for future precision bioelectronic medicine.

Electroceuticals, an emerging class of bioelectronic medicine, utilize implantable devices to target specific organ functions through precise modulation of neural signaling patterns[1–3]. Since unselective neurostimulation often causes undesired side effects (cough, hoarseness, voice alteration, anxiety, headache, etc.)[4], there is an urgent demand for the selective modulation of fine peripheral nerves (<300 μm) adjacent to the target organ, such as the pancreatic and splenic nerves[5,6]. This approach offers a promising method for treating refractory autoimmune, cardiovascular, and metabolic diseases while reducing side effects[7–9]. Compared to intraneural or penetrating electrodes, cuff electrodes that wrap around peripheral nerve bundles are less invasive[10–12]. However, since near-organ fine nerves exhibit individualized anatomical variations and are mechanically fragile[13,14], it remains a longstanding challenge for standard-sized commercial cuff electrodes to provide reliable, durable, and non-disruptive manipulation of near-organ fine nerves during chronic neuromodulation[15,16].

Further, cuff electrodes may change their positions due to micromotion after implantation[17]. Although mechanical suturing of cuff electrodes aid in anchoring and positioning, it tends to induce stress compression and exacerbates inflammation[18,19].

To mitigate the high stress compression on peripheral nerves, modified cuff electrodes employing advanced mechanical fixation methods have been explored[20]. Examples include elaborate fastenings with well-designed mechanical structures[19,21–24], or self-healing elastomers[25], and methods to encircle nerves using shape memory polymers[26,27] or mechanically adaptive hydrogels[28]. Nevertheless, these methods are only suitable for thick nerves (diameters in the millimeter range) owing to the better conformability. As nerve size decreases, the complexity of surgical implantation and mechanical fixation increases dramatically, which can cause irreversible damage to the fine and fragile nerves[29]. Moreover, previous fixation methods also fail to maintain intimate electrical coupling over time[24,29,30], primarily due to

[1]National Engineering Research Center for Nanomedicine, College of Life Science and Technology, Huazhong University of Science and Technology, Wuhan 430074, China. [2]Department of Ultrasound Medicine, Union Hospital, Tongji Medical College, Huazhong University of Science and Technology, Wuhan 430022, China. [3]Department of Materials Science and Engineering, University of Pennsylvania, Philadelphia, PA 19104, USA. [4]These authors contributed equally: Ming Yang, Lufang Wang, Wenliang Liu. ✉e-mail: zli429@hust.edu.cn; ywjiang@seas.upenn.edu; zhiqiangluo@hust.edu.cn

the weak adhesion between the device and the tissue[25,28]. To address these challenges, bio-adhesive conducting hydrogels have been explored for constructing robust neural interfaces[31–35]. However, previous works have been limited to larger tissues such as rat sciatic nerves (~2 mm)[31–33] and heart surfaces[34,35]. Hydrogel bioelectronics for fine nerves (<300 μm) remains elusive.

We envision that a combination of bio-adhesive conducting hydrogels and cuff electrodes can construct robust neural interfaces for fine nerves. In this approach, cuff electrodes, with diameters much larger than nerves, can be loosely sutured to avoid stress compression, while bio-adhesive hydrogels can subsequently fill the gap between the cuff electrode and the nerves. To this end, the hydrogel needs to be (1) injectable to make the cuff fit fine nerves of any shapes and sizes with reduced operational complexity; (2) mechanically stable and anti-swelling to ensure seamless tissue integration and minimal damages; and (3) highly conductive to enable effective bidirectional communication for electrical recording and stimulation. Existing injectable conducting hydrogels, which are mostly made of dynamic reversible networks mixed with conductive fillers, suffer from poor durability and conductivity due to weak molecular interactions and uneven electrical percolation[36,37]. Although incorporating strong irreversible covalent bonds can enhance the mechanical stability, it tends to compromise the injectability[38,39].

Herein, we propose a strategy where a molecular regulator could be introduced during the formation of hydrogel networks to fine tune the reaction kinetics. Typically, based on a biocompatible click chemistry reaction between thiol and maleimide, we discovered that tannic acid (TA) with abundant phenol groups can realize competitive

binding with thiols and effectively modulate the gelation kinetics for optimal injectability while retaining excellent mechanical stability (Fig. 1(i)). Additionally, TA can also enhance multi-scale hydrogen bonding interactions between micron-scale electrically conducting MXene and nano-scale conducting polymer PEDOT:PSS for enhanced conductivity (Fig. 1(ii)). Furthermore, TA bridges between the hydrogel matrix and the conductive network, rendering the hydrogel excellent anti-swelling property (Fig. 1(iii)). Finally, the rich crosslinking chemistries within the network offers instant bio-adhesion (Fig. 1(iv)), allowing for highly conformal and effective electrical coupling with fine nerves. In conjunction with cuff electrodes, the injectable, conductive, adhesive, anti-swelling (ICAA) hydrogel can fill the gap between the electrodes and nerve tissues to establish form-fitting ICAA-cuff (ICAA-C) neural interfaces (Fig. 1, left). Using the rat vagus nerve as a model system for fine peripheral nerves, we showed that chronic neuromodulation using the ICAA-C neural interface in post-myocardial infarction therapy can reduce inflammation, inhibit sympathetic nerve activity, and decrease myocardial fibrosis, thereby maintaining heart function. Overall, our injectable multifunctional hydrogel bioelectronics offer promising opportunities to target various challenging anatomical locations, paving the way toward precision neuromodulation for bioelectronic medicine applications.

## Results
### Injectability, mechanical, and anti-welling properties of ICAA hydrogel
Self-elimination or degradation of typical injectable hydrogel with reversible covalent bonds[40–42] or non-covalent bonds[43–45] is beneficial

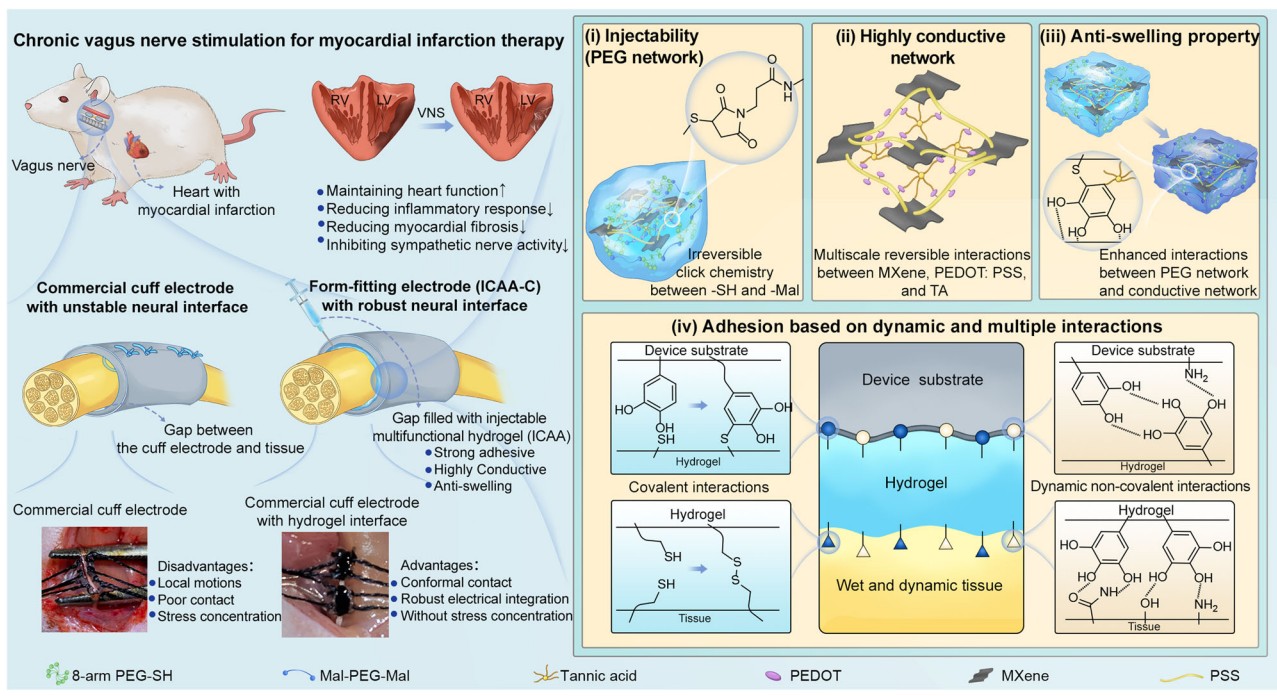

**Fig. 1 | Schematic illustration of the form-fitting cuff electrode (ICAA-C) enabled by the ICAA hydrogel for chronic vagus nerve stimulation.** Form-fitting ICAA-C neural interface can be constructed by injecting ICAA hydrogel to fill the gap between large commercial cuff electrodes and fine nerves. Compared to smaller commercial cuff electrodes with inner diameter close to nerves that fixed with mechanical sutures, the ICAA-C enables conformal contact, robust electrical integration without stress concentration. The ICAA-C can realize long-term, reliable, effective, and safe stimulation of fine rat vagus nerve for the treatment of myocardial infarction. The ICAA hydrogel, consisting of multifunctional PEG network and multi-scale conductive network, is prepared by introducing tannic acid (TA) as a molecular regulator to control the reaction kinetics of click chemistry within PEG network and enhance the multi-scale interactions within conductive

network: (i) The multifunctional PEG network, which is constructed by controllable click reaction kinetics between self-prepared PEG-8SH (Supplementary Fig. 1), PEG-2Mal and TA, facilitates the injectability and mechanical stability of ICAA hydrogel. (ii) The multi-scale conductive network constructed by reversible non-covalent bonds, is created by combining self-prepared micron-scale MXene sheets (Supplementary Fig. 2) with nano-scale PEDOT:PSS dispersion, and molecular regulator TA which enhances the interaction between MXene and PEDOT:PSS. (iii) TA can enhance interactions between PEG network and conductive network, thus renders ICAA hydrogel anti-swelling property (Fig. 1(iii)), which is beneficial to its electrical and mechanical durability. (iv) ICAA hydrogel can easily realize adhesion with both tissues and device substrates by a combination adhesion mechanism based on click chemistry and multiple dynamic non-covalent interactions.

for transient applications, such as drug release or tissue engineering. However, for electroceutical applications, the long-term interfacial stability of hydrogel bioelectronics is crucial. Strong covalent bonds formed by click chemistry between thiols (-SH) and maleimides (-Mal) can ensure the stability of hydrogels, but they suffer from rapid reaction kinetics. For example, in the physiological pH at ~7, the gelation time between 8-arm PEG-thiol (PEG-8SH) and maleimide-PEG-maleimide (PEG-2Mal) is less than 10 s (Fig. 2a), which renders the system unsuitable for injection[46]. In our ICAA hydrogels, to slow down the reaction kinetics of -SH based click chemistry under neutral pH, we introduced TA as the molecular modulator whose abundant phenol groups can competitively react with the -SH in PEG-8SH (Fig. 2a). As a result, the ICAA hydrogel can be easily injected as a homogenous mixture onto any random surfaces, where the conformal and intimate contacts remained stable even under large deformation (Supplementary Fig. 3).

We investigated the injectability, mechanical properties, and stability of our ICAA hydrogels in detail by preparing samples with varied solid contents of PEG-8SH, PEG-2Mal, and MXene (ICAA (15 wt%, 10 wt%, and 4 wt%); ICAA-1 (5 wt%, 3.33 wt%, and 1.33 wt%); ICAA-2 (10 wt%, 6.6 wt%, and 2.66 wt%)) (Supplementary Table 1). As the solid content of the ICAA precursor increased, the gelation time decreased, and the mechanical modulus increased. This is attributed to the increase in cross-linking density within the ICAA hydrogel (Fig. 2b and Supplementary Fig. 4a). Further increasing the solid content resulted in high viscosity and fast gelation, which may impair the injectability. We further investigated the influence of environmental variables on gelation time of ICAA hydrogel, including temperature, pH levels, ionic strength (Supplementary Fig. 4b–d). It turns out that the increase of temperature and pH will shorten the gelation time, due to the accelerated reaction kinetics between the sulfhydryl group and maleimide group under higher temperature and alkaline condition. Moreover, increase of ionic strength can shield the electrostatic interactions between PEDOT and PSS, induce phase separation, and promote the gelation of PEDOT: PSS, thus also greatly shorten the gelation time of ICAA hydrogel.

Most conventional electrode materials exhibit orders of magnitude higher moduli (typically 1 MPa to 1 GPa) than that of soft physiological tissues (<100 kPa, typical elastic moduli of nerve tissues range from a few 100 Pa to about 10 kPa) and are not sufficient to provide mechanical matching interfaces[47,48]. Moreover, soft physiological tissues are always in a dynamic mechanical environment (with strain up to 20% and frequencies from 1 to 10 Hz), bearing shear, tensile and compressive forces[35]. Therefore, the mechanical properties of the hydrogels under shear, tensile and compressive forces were detailed investigated. The G' of the ICAA hydrogels were invariably higher than their G'' and maintain stable across the angular frequency range of 0.5–100 rad s$^{-1}$, indicating the ICAA hydrogel can maintain stable under dynamic conditions (Supplementary Fig. 5a). The ICAA hydrogel displayed a Young's modulus ranging from 6.42 to 40.9 kPa (Supplementary Fig. 5b). Tensile tests of ICAA hydrogel revealed an increase of ~1.8 times for elongation at break and change of tensile modulus from $10 \pm 2$ to $32.3 \pm 2.9$ kPa following an increase in the solid content of hydrogels (Fig. 2c, e). The same trends were also observed in compressive measurements for the strength of the ICAA hydrogel and the compressive moduli (from 42.7 to 149 kPa) when increasing the solid contents (Fig. 2d and Supplementary Fig. 6). These results indicate that the modulus and deformation of ICAA hydrogel can match that of soft biological tissues, especially nerve tissues even under dynamic physiological environment.

The ICAA hydrogel showed minimal swelling and reached equilibrium within 24 h (Fig. 2f). Increasing the solid content resulted in an increase in the crosslinking density, which in turn caused a reduction of equilibrium swelling ratio from $23.46 \pm 2.17\%$ (ICAA-2 hydrogel) to $9.22 \pm 0.76\%$ (ICAA hydrogel) (Supplementary Fig. 7a). For comparison, a pure PEG hydrogel with high solid content (15% PEG-8SH and 10% PEG-2mal) exhibited pronounced swelling in 24 h, reaching equilibrium after 72 h with a swelling ratio of $20.75 \pm 1.1\%$ (Supplementary Fig. 7b). The introduction of MXene/PEDOT:PSS conductive network reduced the equilibrium swelling ratio to $16.04 \pm 1.51\%$. Notably, the TA-enhanced interaction between the PEG network and MXene/PEDOT:PSS network further reduced the swelling ratio of the ICAA hydrogel to $9.22 \pm 0.76\%$ (Supplementary Fig. 7c). The ICAA hydrogel exhibited insignificant mass loss and volume change even after a 4-week immersion in PBS at 37 °C, demonstrating its strong stability (Fig. 2g and Supplementary Fig. 8a, b). More importantly, the ICAA hydrogel exhibited a stable Young's modulus during 4-week test under dynamic loading and body temperature (pH = 7.4 or pH = 6), indicating its long-term mechanical durability (Supplementary Fig. 8c, d).

Compared to the equilibrium swelling rate, gelation time is more crucial for the implantation feasibility of the hydrogel. In contrast, the equilibrium swelling rate and structural stability are more critical for its long-term safety. An appropriate gelation time provides operators with sufficient time to complete the injection and positioning without issues caused by rapid gelation. This ensures the hydrogel is evenly distributed at the target site and remains stable, maintaining its structure and function. Additionally, the equilibrium swelling rate and structural stability accurately reflect the hydrogel's stability after implantation. Although the ICAA hydrogel takes time to reach swelling equilibrium, its adjustable gelation time (42.4 s to 252.1 s), low equilibrium swelling rate, and structural stability ensure both procedural feasibility and post-implantation safety.

## Adhesion properties of ICAA hydrogel

Tissue adhesive mechanisms, based on reactive groups, such as polyphenols[49,50], aldehydes[51,52], and N-hydroxysuccinimide esters[53,54], have been widely used in the preparation of adhesive hydrogels. However, these methods are often unsuitable for enhancing the adhesion of injectable hydrogel bioelectronics given the requirements of long-term mechanical stability and injectability. Applying these adhesion mechanisms to device substrates also poses significant challenges. To address these issues, we proposed a combination adhesion mechanism based on click chemistry and multiple dynamic non-covalent interactions, which can easily facilitate adhesion with both tissues and device substrates.

Spatiotemporally controlled adhesion of the ICAA hydrogels can be accomplished during the sol-gel conversion by tuning the reaction kinetics of -SH based click chemistry within designed gelation time (Fig. 2h). Fluidic ICAA precursor can be injected to the wet device substrate (with PDA coating) or to the tissue surface. In the sol state, the precursor conformally interlocked with the contacting surface by rapidly absorbing and removing interfacial water, forming multiple dynamic non-covalent interactions, such as hydrogen bonds, due to the rich phenol groups within TA. During the sol-gel transition phase (~60 s), a continuous SH-based click reaction takes place within the diffusing ICAA precursors at the hydrogel-substrate or hydrogel-tissue interface. This reaction is the primary driving force allowing the ICAA hydrogel to be firmly anchored between the wet device substrate and tissue. The ICAA hydrogel can maintain its adhesion under continuous washing with physiological buffer, and even when immersed in physiological buffer after 24 h (Supplementary Fig. 9). In the gel state, ICAA hydrogels possess dense cross-linking that restricts the diffusion of polymers and feature a smooth wet surface. This property helps to avoid unwanted adhesion to surrounding tissues and reduces the likelihood of post-operative adhesions (Supplementary Fig. 10).

To quantitatively analyze the adhesion strength and interfacial toughness of ICAA hydrogel on nerve tissues and PDA-coated device substrates, peeling and lap-shear tests were conducted. The interfacial toughness and adhesion strength of the ICAA hydrogel with nerve

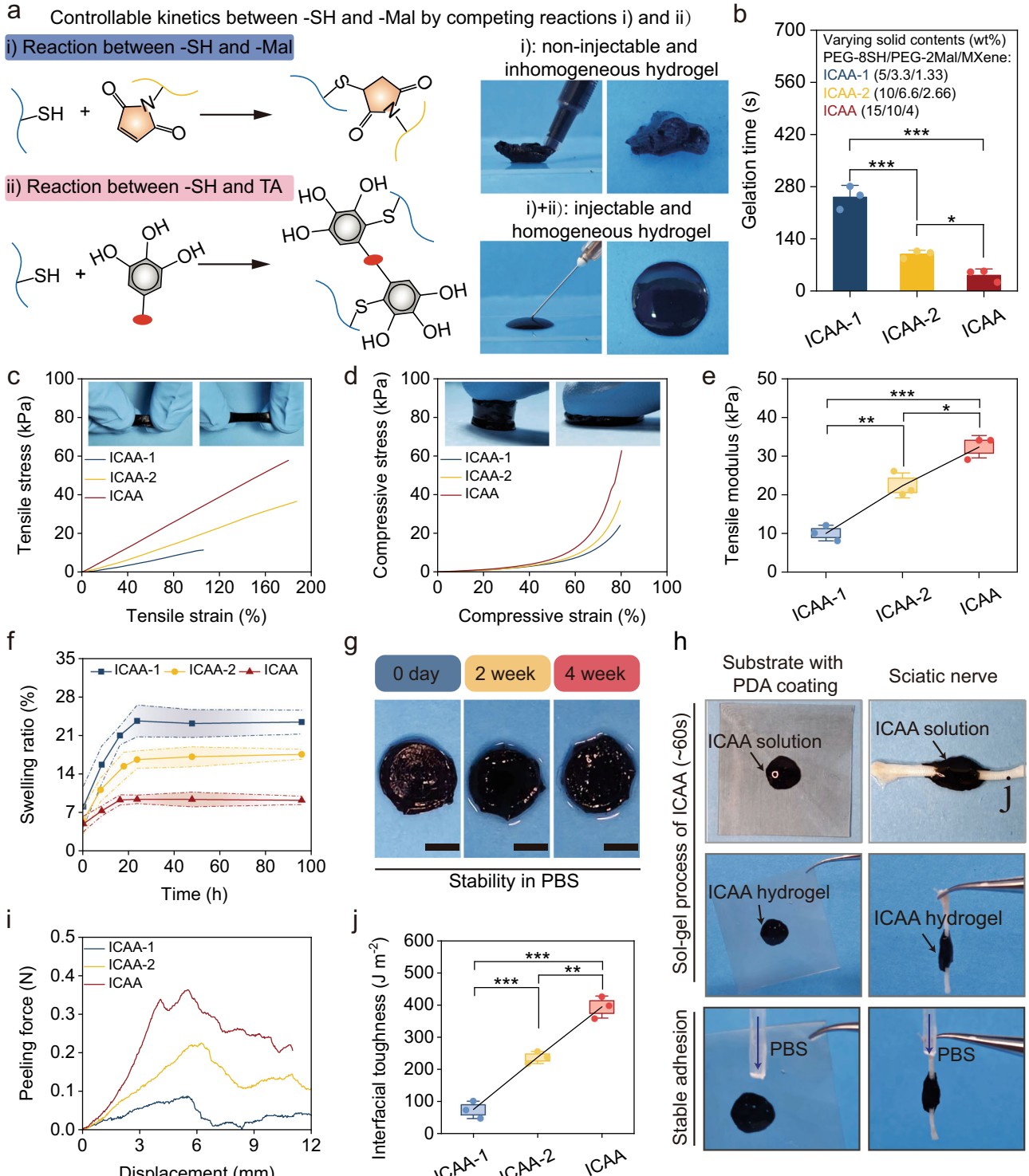

**Fig. 2 | Injectability, stability, mechanical and adhesive properties of ICAA hydrogel. a** Schematic illustration of the controllable reaction kinetics between -SH and -Mal by introducing competing reactions: (i) -SH and -Mal and (ii) -SH and TA. **b** Quantitative analysis of gelation time of ICAA hydrogel suggest its desirable injectability ($n = 3$ independent experiments). **c** Representative tensile stress-strain curves of the ICAA hydrogels. **d** Representative compressive stress-strain curves of the ICAA hydrogels. **e** Quantitative analysis of tensile modulus of the ICAA hydrogels ($n = 3$ independent experiments). **f** Quantitative analysis of the ICAA hydrogel's swelling ratio over time in PBS buffer at 37 °C ($n = 3$ independent experiments). **g** Photographs of the ICAA hydrogel immersed in PBS during 4 weeks. Scale bar, 5 mm. **h** Instantaneous adhesion of the ICAA hydrogel on a silicone with PDA coating and sciatic nerve. **i, j** Representative load-displacement curves for ICAA hydrogel-tissue interface in peeling measurements (**i**), and the interfacial toughness between ICAA hydrogels and nerve tissues (**j**) ($n = 3$ independent experiments). ICAA hydrogels are prepared with varying solid contents of PEG-8SH, PEG-2Mal, and MXene (ICAA (15 wt%, 10 wt%, and 4 wt%); ICAA-1 (5 wt%, 3.33 wt%, and 1.33 wt%); ICAA-2 (10 wt%, 6.6 wt%, and 2.66 wt%)). Data are presented as the mean ± standard deviation in (**b**, **e**, **f**, **j**) and were analyzed by one-way ANOVA first, and then by the Tukey's post hoc test in (**b**, **e**, **j**). *$p \leq 0.05$, **$p \leq 0.01$, ***$p \leq 0.001$. **b** $p = 2.94 \times 10^{-4}$ (ICAA-2 vs ICAA-1), $p = 4.94 \times 10^{-5}$ (ICAA vs ICAA-1), $p = 0.0409$ (ICAA vs ICAA-2). **e** $p = 3.67 \times 10^{-3}$ (ICAA-2 vs ICAA-1), $p = 1.47 \times 10^{-4}$ (ICAA vs ICAA-1), $p = 0.0102$ (ICAA vs ICAA-2). **j** $p = 8.37 \times 10^{-4}$ (ICAA-2 vs ICAA-1), $p = 1.83 \times 10^{-5}$ (ICAA vs ICAA-1), $p = 1.04 \times 10^{-3}$ (ICAA vs ICAA-2).

tissues improved significantly with an increase in the solid contents (Fig. 2i and Supplementary Fig. 11a). This enhancement is attributed to the increased cross-linking density, which enhances the hydrogel's capability to keep interfacial interlocking and confine cohesive failure. Consequently, the toughness and the adhesive strength of ICAA with nerve tissue increased by 5.3 times and 1.89 times in contrast to ICAA-1 hydrogel, respectively (Fig. 2j and Supplementary Fig. 11b). Moreover, ICAA hydrogel also showed strong adhesion with PDA-coated substrate, and its toughness and the adhesive strength were up to $588.6 \pm 31.6 \, J \, m^{-2}$ and $15.6 \pm 1.9 \, kPa$, respectively (Supplementary Fig. 12).

## Electrical and electrochemical properties of ICAA hydrogel

The majority of previously reported injectable conductive hydrogels were prepared by simple blending of conductive additives with the hydrogel network where inhomogeneous electrical percolation often leads to poor electrical durability[36]. In this work, we introduced self-assembled MXene and PEDOT:PSS as the conductive additives, where the former one possesses a lamellar structure (~25 μm) and the latter one is a water-dispersible colloidal suspension (~100 nm). Both materials can stack and entangle themselves into a conductive network by π−π stacking and hydrogen bond interactions. Additionally, TA can serve as a molecular regulator to introduce abundant hydrogen bonding interactions, which facilitate the π-π stacking between PEDOT$^+$ chains and MXene (Fig. 3a). As a result, compared to simple mixing of MXene and PEDOT:PSS, the non-covalent multi-scale conductive network strategy enabled an enhanced conductivity.

To elucidate the multi-scale self-assembly mechanism among MXene, PEDOT:PSS, and TA, Raman spectroscopy and X-ray photoelectron spectroscopy (XPS) characterizations were performed. In the Raman spectrum of PEDOT:PSS film (Fig. 3b), the band at $1430 \, cm^{-1}$ originates from the $C_\alpha = C_\beta$ stretching vibration of thiophene rings. In the MXene/PP film, this band shifted to $1424 \, cm^{-1}$, indicating a conformational change from the benzene to quinoid structure, thereby elongating the conjugation lengths of PEDOT$^+$ chains. Additionally, the narrower bands observed in MXene/ PEDOT:PSS(PP) and MXene/PP/ TA films compared to the PP film confirm increased crystallinity, resulting from the expansion and π−π stacking of PEDOT$^+$ chains[55,56]. Compared to the Ti 2p XPS spectra of MXene, the Ti−C contribution in MXene/PP and MXene/PP/TA increased, as indicated by Supplementary Table 2, suggesting enhanced Ti−C interactions within these composites. The peak in the C 1s XPS spectra corresponding to the C−Ti bond in MXene/PP/TA increased by 16 times compared to MXene (Supplementary Fig. 13), signifying strengthened Ti−C interactions in MXene/PP/TA[55,57]. Atomic force microscopy (AFM) was used to reveal the effects of the enhanced interactions among MXene, PEDOT:PSS, and TA. Height images (Supplementary Fig. 14) showed that the MXene/PP film had a higher roughness and a more aggregated morphology compared to the PP film. In contrast, the MXene/PP/TA films displayed a more uniform morphology, suggesting enhanced interactions between MXene and PEDOT:PSS due to the incorporation of TA. Moreover, TA can facilitate more effective pathways for electron transfer, as verified by AFM current images (Fig. 3d).

The correlation between the conductivity and composition of ICAA hydrogel was further elucidated by preparing samples with varying components (Supplementary Table 3). The conductivity of the ICAA hydrogel (containing 4 wt% MXene and 0.3 wt% TA) was $92.43 \pm 7.65 \, S \, m^{-1}$. This value significantly exceeded those of MXene hydrogel ($3.35 \pm 0.51 \, S \, m^{-1}$) and MXene/PP hydrogel ($10.56 \pm 1.4 \, S \, m^{-1}$), indicating the highly conductive network formed within MXene/PP/TA (Fig. 3e). The conductivity gradually increased with the increasing content of MXene and TA until it saturated (MXene > 3 wt%, TA > 0.3 wt%) (Supplementary Fig. 15). The ICAA hydrogels demonstrated stable conductivity over a 30-day immersion in PBS at 37 °C (Fig. 3f), which was much higher than the

biological tissue conductivity ($0.3–0.7 \, S \, m^{-1}$). The ICAA, MXene, and MXene/PP hydrogels exhibited lower charge transfer resistance compared to Au electrodes, which are typically used in bioelectronic devices. Among these, the ICAA hydrogel demonstrated the lowest impedance, owing to its interconnected, highly conductive network (Fig. 3g, h).

To assess its neuromodulation capability, the charge injection capability (CIC) of the ICAA hydrogel were further characterized. Under electrical stimulation with short biphasic pulses (1.5 ms), the ICAA hydrogel exhibited a considerably higher CIC of $139 \, \mu C \, cm^{-2}$ compared to Au electrodes ($19.1 \, \mu C \, cm^{-2}$). High current densities of approximately $-2.5 \, mA \, cm^{-2}$ and $6.5 \, mA \, cm^{-2}$ were realized with stimulation voltages as low as $\pm 20 \, mV$ and $\pm 50 \, mV$, respectively (Fig. 3i, and Supplementary Fig. 16a, b), indicating that the ICAA hydrogel significantly promoted the charge injection efficacy. Consequently, the ICAA hydrogel can be used in low pulse-width and low-intensity modes of nerve electrical stimulation for improved biosafety. Stable CIC and interfacial impedance are imperative for reliable and efficient in vivo neuromodulation[58]. The ICAA hydrogel demonstrated an insignificant changes of CIC value under $10^5$ charging and discharging cycles (Fig. 3j and Supplementary Fig. 17a). Moreover, the conductivity of the ICAA hydrogel maintained ~95 S m$^{-1}$, indicating its electrical stability during prolonged stimulating test (Supplementary Fig. 17b). The ICAA hydrogel exhibited stable impedance during a 4-week immersion in PBS (Fig. 3k), maintaining an impedance of approximately 120 Ω at 1 kHz even after 4 weeks (Fig. 3l). Furthermore, the ICAA hydrogel displayed negligible change in its CIC values during a 4-week immersion in PBS (Fig. 3m, n).

## Stable ICAA-C interface for chronic neuromodulation

In comparison with previously reported injectable conducting hydrogels based on dynamic covalent and non-covalent bond interactions[35,59–63], the ICAA hydrogel, featuring a stable irreversible covalent PEG network and a multi-scale non-covalent conductive network, has significantly improved electrochemical properties and tissue-like mechanical properties (Fig. 3o and Supplementary Table 4). For instance, most injectable hydrogels for bioelectronics commonly possess low conductivities ($0.065–12.5 \, S \, m^{-1}$), while ICAA possesses a much-increased conductivity of $92.43 \pm 7.65 \, S \, m^{-1}$. Notably, different from typical injectable conductive hydrogels, the ICAA hydrogel shows anti-swelling and long-term stability to ensure mechanical and electrical durability. Meanwhile, the ICAA hydrogel also demonstrates strong tissue adhesion (20.9 kPa). The decent stability and the ability to meet various performance requirements of hydrogel bioelectronics, make ICAA hydrogel highly advantageous than previous reported injectable hydrogels in neural interface for fine nerves.

The potential of ICAA-C as chronic neural interface was assessed through detailed in vitro and in vivo feasibility studies. The CCK-8 and live-dead staining methods demonstrated that ICAA had little effect on cell proliferation and morphology (Supplementary Fig. 18a, b). The injection procedure of ICAA hydrogel show it can be simply injected by syringe needle (Supplementary Fig. 19a), and its shape demonstrated minor changes right after implantation and 4 weeks post-implantation, indicating its shape stability after implantation (Supplementary Fig. 19b).

Although the goal is to use the neural interface in humans, prior to clinical applications in humans, the effects of neural interface for fine nerves should be assessed in pre-clinical rodent models. In vivo immune response and cell population changes in rats' vagus nerve tissues post-4-week implantation were analyzed using immunofluorescence for Schwann cells (S-100), neurofilaments (NFM), and macrophages (Iba-1). ICAA-C was constructed by injecting ICAA hydrogel to fill the gap between the commercial cuff electrodes (inner diameter of 500 μm) and vagus nerves (diameter of ~300 μm). As a positive control, commercial cuff electrodes (300 μm inner

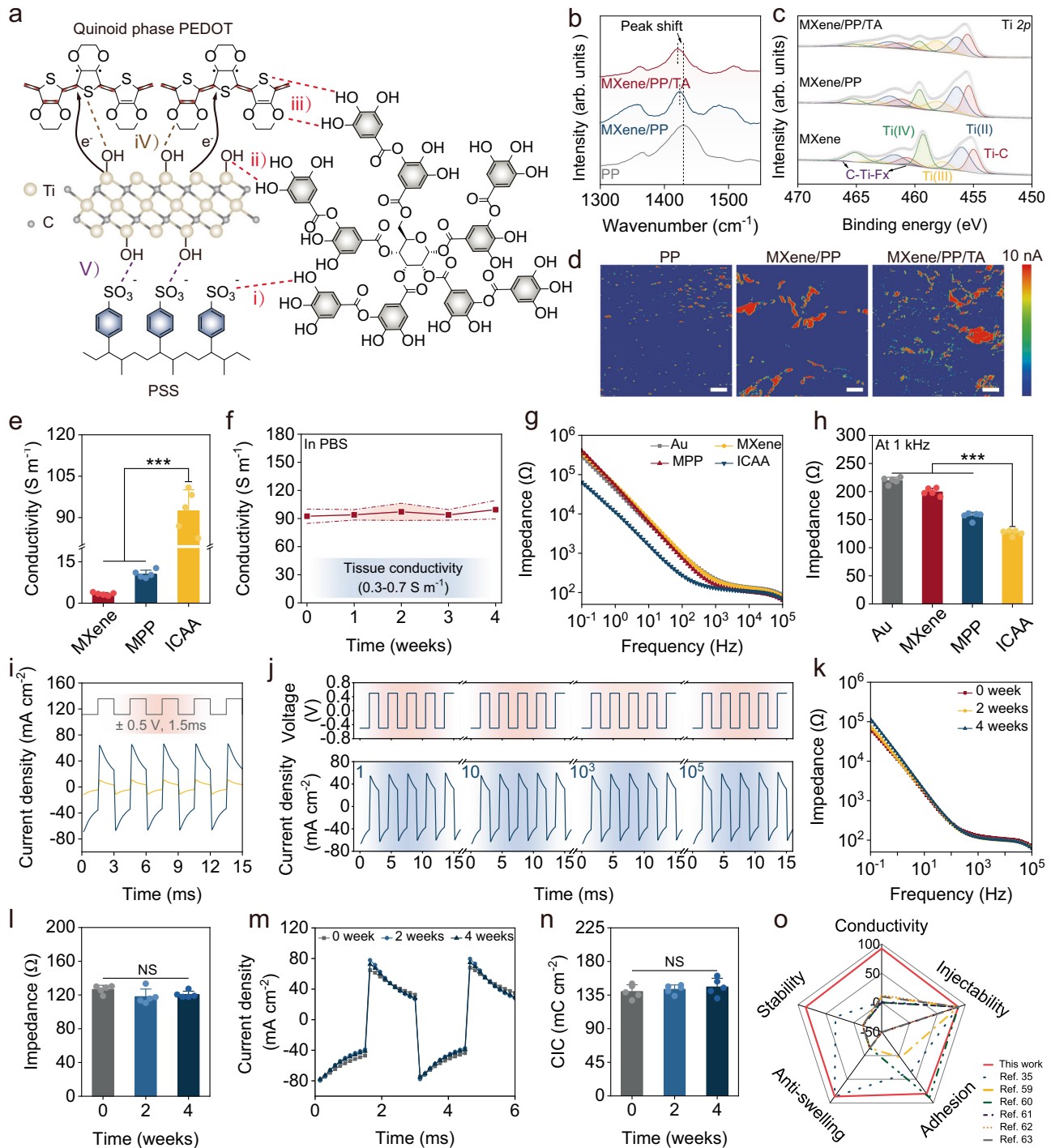

**Fig. 3 | Electrical and electrochemical properties of the ICAA hydrogel.**
**a** Schematic illustration shows multi-scale interactions between MXene, PEDOT:PSS (PP), and TA. (i)–(iii) Hydrogen bond interactions between TA and PEDOT, PSS, MXene. (iv) Hydrogen bond interactions and π–π stacking between MXene and PEDOT. (v) Hydrogen bond interactions between MXene and PSS. **b, c** Raman and XPS spectra compare MXene/PP/TA, MXene/PP, and PP. **d** AFM current images of MXene/PP/TA, MXene/PP, and PP. Scale bar, 250 nm. **e** Conductivities of PEG hydrogels mixed with these conductive phases ($n = 5$ independent experiments). **f** Conductivities of the ICAA hydrogels during 4-week immersion in PBS ($n = 5$ independent experiments). **g** Impedances of PEG hydrogel and ICAA hydrogels. **h** Impedances at 1 kHz of ICAA hydrogels ($n = 5$ independent experiments). **i** CIC curves of ICAA hydrogel (blue) vs. Au electrode (yellow) with biphasic pulses

($\pm 0.5$ V, 1.5 ms). **j** Cycling stability of CIC for ICAA hydrogel over 100,000 cycles. **k**, **l** Impedance of ICAA hydrogel in PBS at 37 °C over 4 weeks ($n = 5$ independent experiments). **m**, **n** CIC curves and stability of ICAA hydrogel in PBS at 37 °C over 4 weeks ($n = 5$ independent experiments). **o** Comparison of ICAA with other reported injectable conductive hydrogels[35,59–63], regarding injection, conductivity, adhesion, anti-swelling, and stability. Data are presented as the mean ± standard deviation in (**e, f, h, l, n**) and were analyzed by one-way ANOVA first in (**e, h, l, n**), and then by the Tukey's post hoc test. NS, not significant, ***$p \leq 0.001$. **e** $p = 0.0632$ (MPP vs MXene), $p = 0$ (ICAA vs MXene and MPP). **h** $p = 0$ (ICAA vs Au and MXene), $p = 4.14 \times 10^{-6}$ (ICAA vs MPP). **l** $p = 0.12$ (2 weeks vs 0 weeks), $p = 0.3003$ (4 weeks vs 0 week), $p = 0.8238$ (4 weeks vs 2 weeks). **n** $p = 0.8659$ (2 weeks vs 0 weeks), $p = 0.5509$ (4 weeks vs 0 week), $p = 0.8467$ (4 weeks vs 2 weeks).

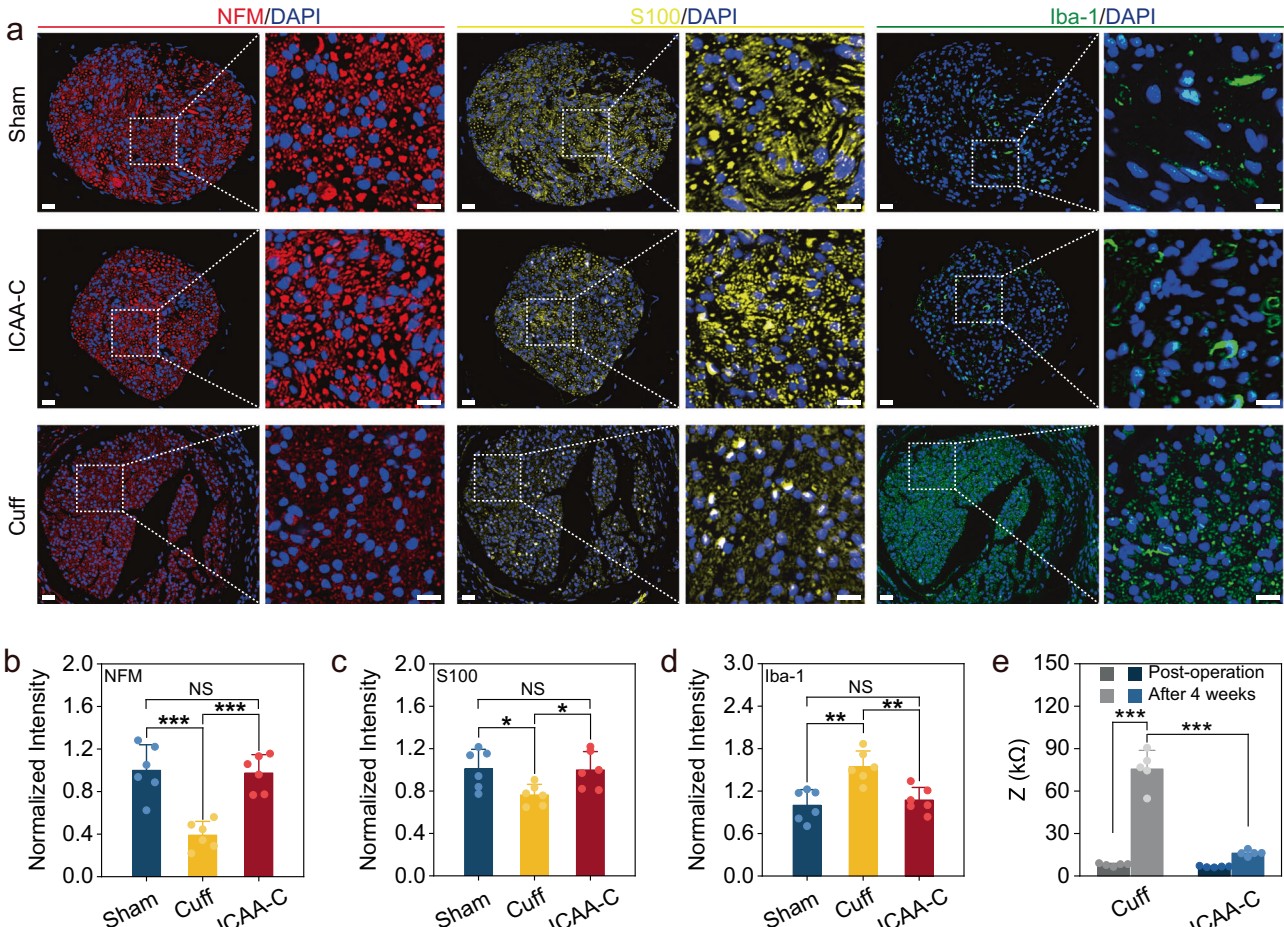

**Fig. 4 | Stable neural interface constructed by ICAA-C. a** Representative immunofluorescence photographs of the vagus nerves at 4 weeks post-implantation in ICAA-C, commercial cuff (diameter of 300 μm), and sham group. The images are magnified ×45 (left), ×175 (right) to show the white dotted regions of the nerves. Scale bar, 20 μm (left), 10 μm (right). **b–d** Normalized fluorescence intensity of neurofilaments (NFM) (**b**), Schwann cells (S-100) (**c**), and macrophages (Iba-1) (**d**) in ICAA-C, commercial cuff (300 μm), and sham group, respectively (*n* = 6 independent animals). ICAA-C implantation shows reduced tissue damage with less inflammatory response in contrast to commercial cuff implantation. **e** Impedances at 1 kHz of the ICAA-C and commercial cuff electrode (diameter of 300 μm) post-

operation and 4 weeks after implantation (*n* = 5 independent animals). ICAA-C shows minimal impedance change, unlike the commercial cuff electrode group, which exhibits a ~9.5-fold increase post-implantation. Data are presented as the mean ± standard deviation in (**b–e**) and were analyzed by one-way ANOVA first, and then by the Tukey's post hoc test. *$p \leq 0.05$, **$p \leq 0.01$, ***$p \leq 0.001$. NS not significant. **b** $p = 1.25 \times 10^{-4}$ (Cuff vs Sham), $p = 0.9703$ (ICAA-C vs Sham), $p = 1.93 \times 10^{-4}$ (ICAA-C vs Cuff). **c** $p = 0.0395$ (Cuff vs Sham), $p = 0.992$ (ICAA-C vs Sham), $p = 0.0495$ (ICAA-C vs Cuff). **d** $p = 1.05 \times 10^{-3}$ (Cuff vs Sham), $p = 0.8123$ (ICAA-C vs Sham), $p = 3.57 \times 10^{-3}$ (ICAA-C vs Cuff). **e** $p = 0$ (Cuff (Post-operation) and ICAA-C (After 4 weeks) vs Cuff (After 4 weeks)).

diameter) matching the nerve diameter were implanted and fixed by sutures. The sham group underwent the same surgery without electrode implantation. There was no significant difference in Schwann cell and axon populations between the ICAA-C and control groups (Fig. 4a–c). In contrast, the positive control group showed significant Schwann cell reduction due to nerve damage, primarily caused by size mismatch and suture compression. Individual nerve size variances prevent a perfect fit of standard-sized cuff electrodes, and suture fixation often exacerbates nerve compression. Moreover, no significant difference was observed in Iba-1 expression between the ICAA-C and sham groups, while a severe immune response occurred in the positive control group (Fig. 4a, d). The positive control group also exhibited the highest fibrosis degree (Supplementary Fig. 20). ICAA-C implantation resulted in minimal impedance changes, unlike the positive control group (~9.5-fold increase), potentially leading to device failure, indicating improved long-term biocompatibility for ICAA-C. Additionally, researches on PEDOT in humans demonstrated its advantages over traditional metal electrodes in high-resolution electrocorticography[64,65], supporting the feasibility of PEDOT for human applications.

## ICAA-enabled neuromodulation for myocardial rehabilitation

We used the cervical vagus nerve of rats as a proximal organ nerve model, as its size is similar to that of the near-organ fine nerves in humans, and the chronic neuromodulation ability of ICAA-C was verified by the rehabilitation therapy after rat myocardial infarction. The biphasic charge-balanced rectangular current pulses (10 Hz, 0.1 mA, 1.5 ms) were applied to stimulate the vagus nerve, following the methodology of previous studies[66,67]. We observed no significant changes in the rats' weight or heart rate during the vagus nerve stimulation treatment (Supplementary Fig. 21). Additionally, typical side-effects such as coughing, hoarseness, or alterations in voice were absent. These results align with previous studies that utilized low-dose vagus nerve stimulation[66,67], likely attributable to our employment of low-dose stimulation. After the 4-week rehabilitation therapy, cardiac morphological staining was employed to assess ICAA-C's therapeutic effects on myocardial repair. H&E staining revealed mild inflammation and immune responses in the ICAA-C group when compared to the MI group, characterized by myocardial cell necrosis, loose tissue and increased transparency (Supplementary Fig. 22). Masson staining analysis of rats in the MI group displayed severe ventricular

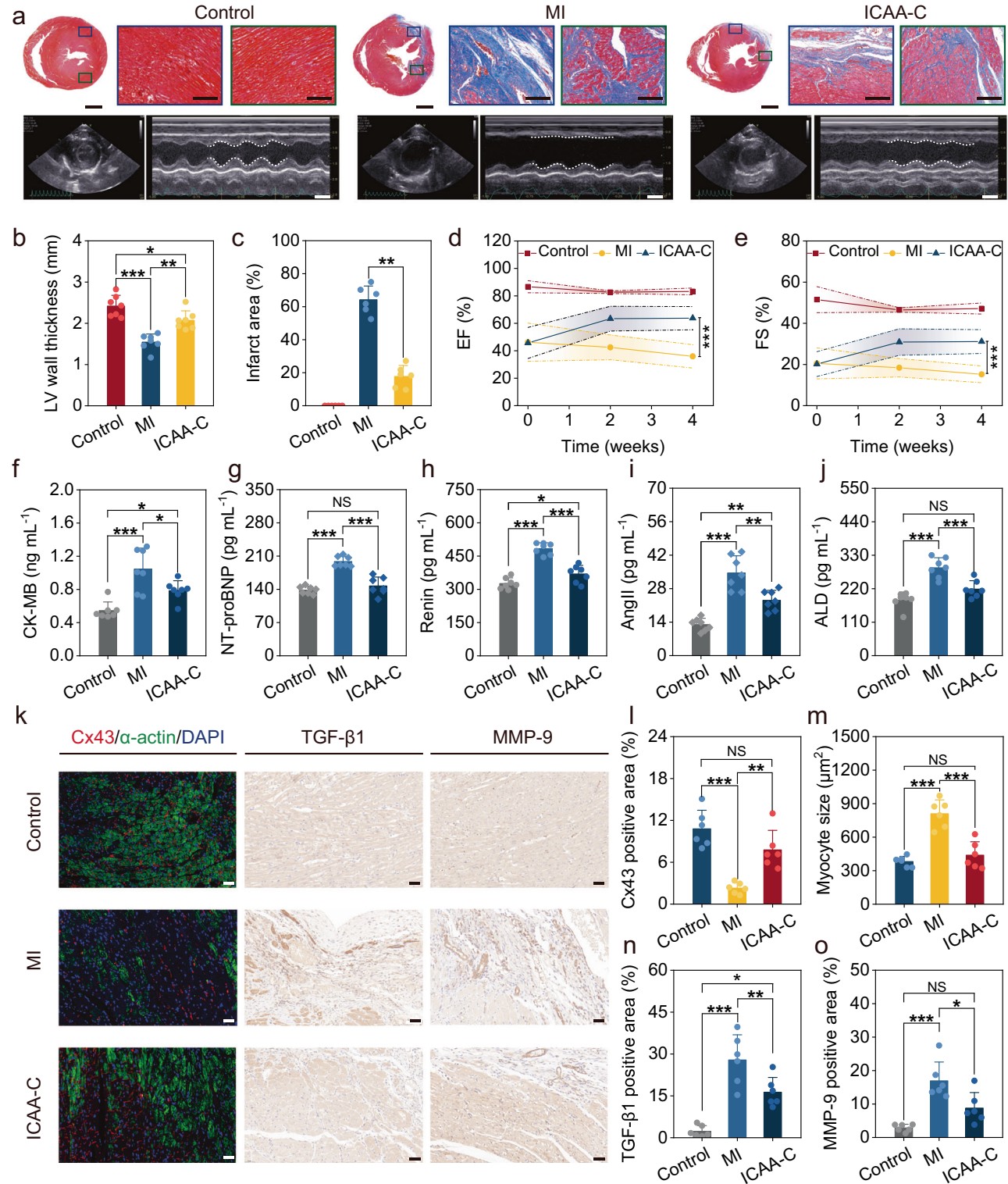

remodeling, characterized by left ventricular (LV) wall thinning, cardiomyocyte apoptosis, and fibrosis deposition, in comparison with the control group (Fig. 5a). On the contrary, ICAA-C-based vagus nerve stimulation suppressed ventricular remodeling by significantly alleviating its thinning (Fig. 5b), reduced myocardial fiber loss and collagen fibrosis deposition, and decreased the MI area from 64.3 ± 8.3% to 17.8 ± 6.7% (Fig. 5c).

Additionally, echocardiography was employed to assess cardiac geometry and function in the ICAA-C, MI, and control groups at 1 day, 2 weeks, and 4 weeks postoperatively (Fig. 5a and Supplementary

Fig. 23a, b). Rats in MI group showed significant ventricular dilatation, evidenced by an increase in left ventricular internal diameter and volume at both end-diastole (LVIDd and EDV) and end-systole (LVIDs and ESV). Consequently, it would lead to a reduction in ejection fraction (EF), fractional shortening (FS), and stroke volume (SV). Conversely, ICAA-C-based vagus nerve stimulation reduced LVIDd and LVIDs, mitigating ventricular remodeling (Supplementary Figs. 24 and 25). Moreover, cardiac function declined after 4 weeks in the MI group, while ICAA-C-based vagus nerve stimulation improved heart beating function. For instance, rats in the ICAA-C group exhibited

**Fig. 5 | The ICAA-C-based chronic vagus nerve stimulation allows for effective MI therapy. a** Masson staining shows collagen deposition (blue), with echocardiography images highlighting left ventricle (LV) size changes at the papillary muscle level during systole and diastole. Scale bar, 2 mm, and 200 μm for Masson staining, and 5 mm for echocardiography, respectively. **b, c** Quantitative analysis of LV wall thickness (**b**) ($n = 7$ independent animals) and infarcted area (**c**) ($n = 6$ independent animals). **d, e** Ejection fraction (EF) (**d**) and fraction shortening (FS) (**e**) of LV at 1 day, 2 weeks, and 4 weeks post-operation ($n = 7$ independent animals). **f–j** Serum level of creatine kinase-MB (CK-MB) (**f**), N-terminal pronatriuretic peptide (NT-proBNP) (**g**), Renin (**h**), angiotensin II (Ang-II) (**i**), aldosterone (ALD) (**j**) at 4 weeks post-operation ($n = 7$ independent animals). **k** Immunofluorescence and immunohistochemical images for cardiac markers (α-actin), Connexin 43 (Cx43), transforming growth factor beta 1 (TGF-β1) and matrix metallopeptidase 9 (MMP-9). Scale bar, 50 μm. **l–o** Quantitative analysis of Cx43 (**l**), myocyte size (**m**), TGF-β1 (**n**), and MMP-9 (**o**) ($n = 7$ independent animals). Data are presented as the mean ± standard deviation and were analyzed by one-way ANOVA first, and then by the Tukey's post hoc test. *$p \leq 0.05$, **$p \leq 0.01$, ***$p \leq 0.001$. NS not significant. $p_1$ (MI vs Control), $p_2$ (ICAA-C vs Control), $p_3$ (ICAA-C vs MI) **b** $p_1$ ($3.43 \times 10^{-6}$), $p_2$ (0.02878), $p_3$ (0.00117). **c** $p_3$ (0.0015). **d** $p_3$ ($2.14 \times 10^{-6}$). **e** $p_3$ ($6.24 \times 10^{-6}$). **f** $p_1$ ($1.04 \times 10^{-4}$), $p_2$ (0.0441), $p_3$ (0.0292). **g** $p_1$ ($2.47 \times 10^{-7}$), $p_2$ (0.4393), $p_3$ ($2.59 \times 10^{-6}$). **h** $p_1$ ($1.02 \times 10^{-8}$), $p_2$ (0.0414), $p_3$ ($4.48 \times 10^{-6}$). **i** $p_1$ ($5.92 \times 10^{-7}$), $p_2$ (0.00363), $p_3$ (0.00125). **j** $p_1$ ($5.82 \times 10^{-6}$), $p_2$ (0.0769), $p_3$ ($7.26 \times 10^{-4}$). **l** $p_1$ ($2.86 \times 10^{-5}$), $p_2$ (0.0858), $p_3$ (0.00215). **m** $p_1$ ($8.36 \times 10^{-6}$), $p_2$ (0.6128), $p_3$ ($4.3 \times 10^{-5}$). **n** $p_1$ ($8.20 \times 10^{-6}$), $p_2$ (0.00337), $p_3$ (0.0135). **o** $p_1$ ($1.08 \times 10^{-4}$), $p_2$ (0.0699), $p_3$ (0.012).

enhanced EF, increasing from 45.7 ± 11.3% to 63.8 ± 8.4% (Fig. 5d), and FS, rising from 20.2 ± 6% to 31.1 ± 5.8% (Fig. 5e), approaching levels indicative of health.

The levels of markers related to cardiac and heart failureare often tested to assist more accurate MI diagnosis and rehabilitation assessment[68]. Typical biomarkers include creatine kinase-MB (CK-MB) and N-terminal pro b-type natriuretic peptide (NT-proBNP), which are indicative of myocardial injury and cardiac insufficiency, respectively. These two biomarkers were significantly higher in the MI group compared to the control group (Fig. 5f, g). Concurrently, renin, angiotensin II (Ang-II), and aldosterone (ALD) related to heart failure, elevated in the MI group compared to the control group 4 weeks post-MI (Fig. 5h–j). Vagus nerve stimulation enabled by ICAA-C markedly led to the downregulation of these markers by more than 15%, further indicating its therapeutic effect.

The reversal of LV cellular remodeling by ICAA-C-based vagus nerve stimulation was investigated by immunofluorescence and immunohistochemistry of LV tissues. In the MI group, connexin 43 (Cx43) expression in the MI area was diminished, suggesting impaired intercellular electrical communication. ICAA-C-based vagus nerve stimulation augmented Cx43 expression, thereby enhancing electrical signaling and contraction coupling among adjacent cardiomyocytes (Fig. 5k, l). Cardiomyocytes with limited regenerative capacity compensate for the loss in number by increasing in volume in MI group, while the myocardial size in rats from the ICAA-C group exhibited a reduction, decreasing it from 810.35 ± 123.43 μm$^2$ to 439.19 ± 119.59 μm$^2$ (Fig. 5m and Supplementary Fig. 26). Furthermore, ICAA-C-based vagus nerve stimulation reduced the positive area fractions of transforming growth factor beta 1 (TGF-β1) (Fig. 5k, n), which markedly suppressed collagen I and III synthesis in MI area in ICAA-C group (Supplementary Fig. 27). And it reduced the matrix metallopeptidase 9 (MMP-9) expression, which can inhibit collagen degradation in distant infarct areas post-MI[69] to avoid LV cellular remodeling (Fig. 5k, o). Additionally, our ICAA-C method showed no adverse effects on other organs, evidenced by consistent HE staining (Supplementary Fig. 28).

The effect of long-term vagus stimulation on anti-inflammation was then studied. After the 4-week treatment by ICAA-C, the expression of two typical inflammatory factors (TNF-α, IL-6) in serum were markedly low than MI group (Fig. 6a, b), suggesting that the stimulation aids in alleviating inflammatory symptoms, thus decelerating the progression of MI. Moreover, activation of the vagus nerve has been demonstrated to inhibit sympathetic nerve activity[70]. This inhibition is essential for the treatment of MI, wherein sympathetic nerves are excessively activated and reorganized into proinflammatory circuits, contributing to left ventricular (LV) remodeling. The ICAA-C-based vagus nerve stimulation resulted in decreased expression levels of norepinephrine (NE) and epinephrine (EPI) throughout the treatment course, compared to the MI group (Fig. 6c, d and Supplementary Fig. 29). This reduction indicates that sustained enhancement of vagus nerve activities by ICAA-C can inhibit the sympathetic overexcitation caused by MI.

We further investigated the effect of long-term vagus stimulation on inhibiting sympathetic remodeling in rat ventricles post-MI using immunofluorescence. Growth-associated protein 43 (GAP-43) was used to assess sympathetic nerve regeneration, while neurofilament 200 (NF200) and tyrosine hydroxylase (TH) evaluated sympathetic nerve activity and innervation density. Increased expression of GAP-43 in the MI group's left ventricle, right ventricle, and interventricular septum suggests enhanced nerve regeneration in these areas (Fig. 6e, f and Supplementary Fig. 30a). Enhanced nerve regeneration significantly increases the innervation density in the heart, as evidenced by the higher expression of NF200 and TH in the MI group in contrast to the control group, indicating overactivation of sympathetic nerves (Fig. 6e, g, h, and Supplementary Fig. 30b). ICAA-C-based vagus nerve stimulation significantly decrease GAP-43, NF200, and TH expression levels in contrast to MI group, thereby inhibiting sympathetic nerve remodeling (Fig. 6f–h). Overall, chronic stable neuromodulation consistently activates the vagus nerve and enables steady regulation of inflammatory factors and sympathetic nerve activity, which may aid in treating various inflammation and cardiovascular-related diseases.

## Discussion

Implementing reliable, durable, and non-disruptive manipulation during chronic neuromodulation of fragile near-organ fine nerves requires a conformal and intimate neural interface between nerves and devices. Currently, cuff electrodes faced challenges such as modulus mismatches, geometrical incompatibility, and interfacial micromotions, often leading to unstable electrical coupling for fine nerves and potential irreversible nerve damage. In this contribution, we report the injectable and robust hydrogel bioelectronics (multifunctional ICAA hydrogel) by introducing TA as a molecular regulator to control the reaction kinetics of click chemistry within a PEG network and enhance multi-scale interactions within a conductive network. The ICAA hydrogel showed good injectability, high conductivity, strong interface–adhesion, and excellent anti-swelling. Meanwhile, it showed substantially improved mechanical and electrical durability in contrast to previously reported injectable conductive hydrogels. We then fabricated a form-fitting ICAA-C neural interface, which offered bioadhesive fixation approach by injecting ICAA hydrogel between commercial cuff electrodes and fine nerves, replacing the mechanical suture fixation of cuff electrodes to avoid irreversible nerve damage. This approach can enable stable electrical integration between loosely sutured large cuff electrodes and fine nerves. Using the rat vagus nerve as a model of fine peripheral nerves, we demonstrated that the chronic neuromodulation through the ICAA-C neural interface can deliver effective neuromodulation rehabilitation therapy of post-MI, which successfully alleviated inflammatory responses, inhibited sympathetic nerve activity, lessened myocardial fibrosis, and maintained heart functions. In the future, the ICAA-C-based neural interface can be readily adapted for a variety challenging anatomical locations, such as near-organ fine nerves for abdominal electroceuticals, intradural space for spinal cord stimulation, specific dorsal root ganglion for chronic

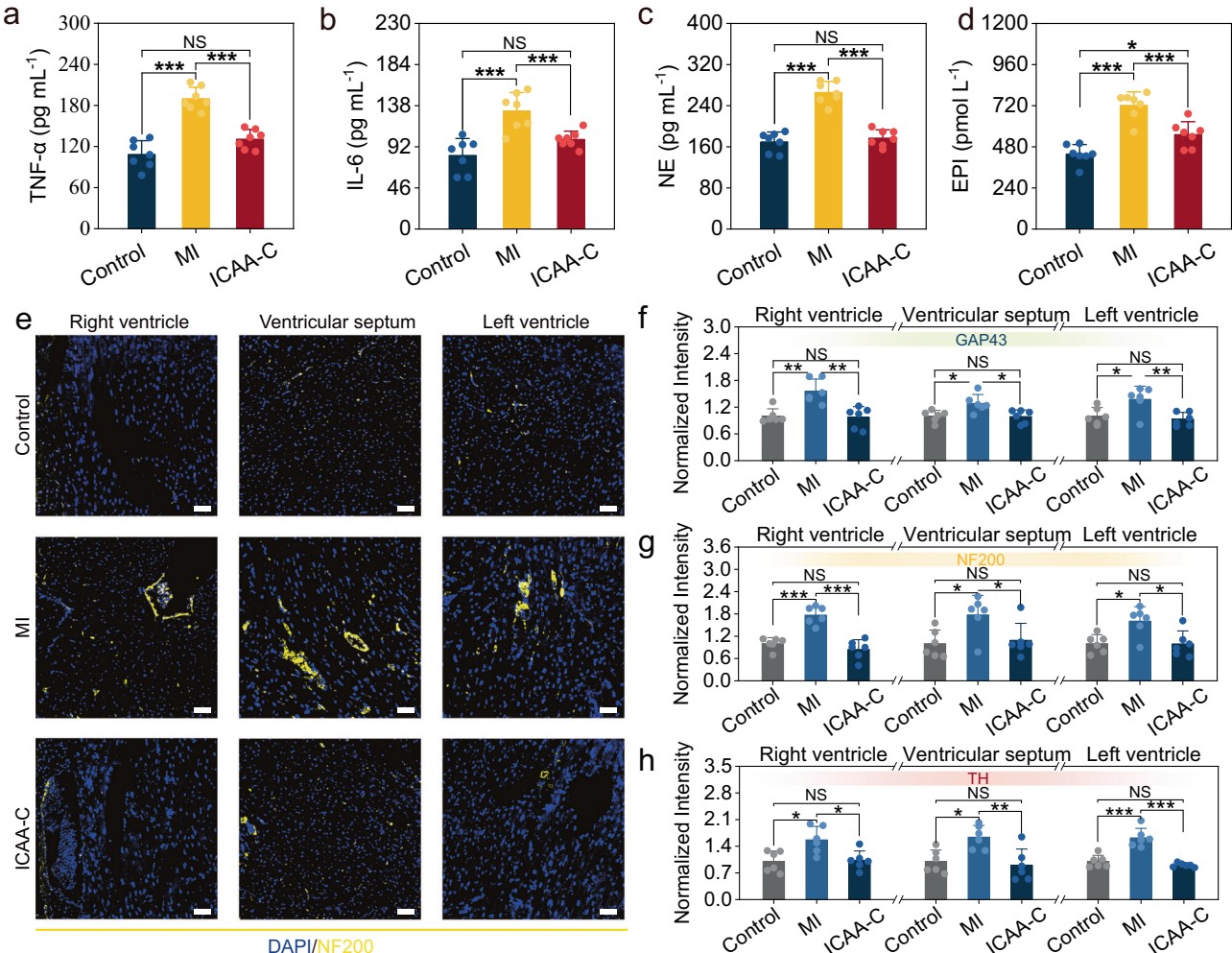

**Fig. 6 | ICAA-C-based vagus nerve stimulation regulates inflammation and cardiac sympathetic nerve activity. a, b** Serum level of tumor necrosis factor-alpha (TNF-α) (**a**), and interleukin-6 (IL-6) (**b**) at 4 weeks post-operation ($n = 7$ independent animals). **c, d** Serum level of norepinephrine (NE) (**c**), and epinephrine (EPI) (**d**) at 4 weeks post-operation ($n = 7$ independent animals).
**e** Immunofluorescence staining of the left ventricle, right ventricle, and ventricular septum of rat heart in the ICAA-C, MI, and control groups for neurofilament 200 (NF200). Scale bar, 50= μm. **f–h** Normalized fluorescence intensity of ICAA-C, MI, and control group respectively for **f** GAP43, **g** NF200, **h** TH ($n = 6$ independent animals). Data are presented as the mean ± standard deviation in (**a–c, d, f–h**), and were analyzed by one-way ANOVA first, and then by the Tukey's post hoc test.

*$p ≤ 0.05$, **$p ≤ 0.01$, ***$p ≤ 0.001$. NS not significant. $p_1$ (MI vs Control), $p_2$ (ICAA-C vs Control), $p_3$ (ICAA-C vs MI). **a** $p_1$ ($1.09 × 10^{-7}$), $p_2$ (0.06127), $p_3$ ($1.21 × 10^{-5}$). **b** $p_1$ ($9.49 × 10^{-5}$), $p_2$ (0.148), $p_3$ ($6.79 × 10^{-3}$). **c** $p_1$ ($1.84 × 10^{-8}$), $p_2$ (0.7558), $p_3$ ($1.39 × 10^{-7}$). **d** $p_1$ ($1.48 × 10^{-6}$), $p_2$ (0.0182), $p_3$ ($7.17 × 10^{-4}$). **f** Right ventricle: $p_1$ (0.00192), $p_2$ (0.9767), $p_3$ (0.00127). Ventricular septum: $p_1$ (0.022), $p_2$ (0.9928), $p_3$ (0.0175). Left ventricle: $p_1$ (0.0254), $p_2$ (0.8433), $p_3$ (0.0084). **g** Right ventricle: $p_1$ ($8.07 × 10^{-5}$), $p_2$ (0.4737), $p_3$ ($1.01 × 10^{-5}$). Ventricular septum: $p_1$ (0.0252), $p_2$ (0.9378), $p_3$ (0.0487). Left ventricle: $p_1$ (0.0167), $p_2$ (0.999), $p_3$ (0.0155). **h** Right ventricle: $p_1$ (0.0119), $p_2$ (0.982), $p_3$ (0.0171). Ventricular septum: $p_1$ (0.0162), $p_2$ (0.8834), $p_3$ (0.00625). Left ventricle: $p_1$ ($1.19 × 10^{-5}$), $p_2$ (0.6060), $p_3$ ($9.82 × 10^{-6}$).

pain treatment. We envision the application of injectable multi-functional ICAA hydrogel can be further extended to bioelectronic regenerative medicine for repairing of various electro-active tissues, including nerves, heart, and skeletal muscles.

## Methods

### Ethical approval statement
All animal experiments were conducted in accordance with guidelines approved by the Institutional Animal Care and Use Committee of Huazhong University of Science and Technology (IACUC Number: 3663).

### Materials
Ti$_3$AlC$_2$ powder (200 mesh) was purchased from Xinxi technology. Hydrochloric acid solution (HCl, 37%), and toluene (≥99%) were purchased from Sinopharm Chemical Reagent. Eight-arm Polyethylene Glycol (8-arm PEG, Mw = 20,000, ≥95%) was purchased from

Adamas life. Lithium fluoride (LiF, 99%), mercaptopropionic (99%), p-toluenesulfonic acid (98.5%), DL-dithiothreitol (DTT, 99%), sodium hydroxide (NaOH, 96%), dichloromethane (99.5%), maleimide PEG maleimide (PEG-2Mal, Mw = 5000), and methyl tert-butyl ether were purchased from Aladdin. PEDOT:PSS (PH1000, 1.1–1.3 wt%) was purchased from Heraeus Electronic Materials. Tannic acid (TA, ACS) was purchased from J&K Scientific. DI water was collected from a Milli-Q water purification system (Millipore). All reagents were used without purification.

### Synthesis of MXene
To synthesize of MXene, 2 g of Ti$_3$AlC$_2$ powder was slowly added to the etching solution consisting of 3.2 g LiF in 40 mL of 9 M HCl. Etching was carried out for 30 h at 40 °C. The obtained dispersion was washed with DI water by centrifuging at 1027$g$ for 6 min and repeating 3 times, and then centrifuging at 4108.7$g$ for 10 min until supernatant pH of ~6. Next, the precipitate was dispersed in water and the resulting

dispersion was continuously shaken by hand for 20 min. The dispersion was centrifuged at 22.1*g* for 3 min and the unetched samples in the lower layer was discarded. The upper layer was further centrifuged at 12074.4 rpm for 1 h to obtain multilayer MXene.

### Synthesis of 8-arm PEG-SH

To synthesize thiol-functionalized 8-arm PEG (PEG-8SH), 8-arm PEG was dried to remove residual water by azeotropic distillation with toluene before use. Under a flow of nitrogen, 8-arm PEG (10 g, 4 mmol OH groups), mercaptopropionic acid (4.24 g, 40 mmol), p-toluenesulfonic acid (22.8 mg, 0.12 mmol), DTT (154 mg, 1 mmol) were dissolved in toluene (100 mL). The reaction mixture was refluxed with Dean-Stark apparatus under stirring for 24 h (Supplementary Fig. 1). Toluene was removed under reduced pressure and the polymer was dissolved in dichloromethane and precipitated 3 times in cold methyl tert-butyl ether. After drying in vacuo, white solid was recovered and stored at −20 °C for future use.

### Preparation of the ICAA hydrogel

The prepared MXene (0–4 wt%) was redispersed in a PEDOT:PSS suspension, followed by the addition of TA (0.1–0.5 wt%) to the MXene/PEDOT:PSS mixture. The resulting MXene/PEDOT/TA mixture was subjected to magnetic stirring for 2 h, after which the pH was adjusted to 7.4 for subsequent use. PEG-8SH (5–15 wt%) and PEG-2Mal (3.3–10 wt%) were respectively dissolved in the MXene/PP/TA dispersion. ICAA hydrogels were prepared by combining equal volumes of the prepared PEG-8SH and PEG-2Mal solutions using a connecting device.

### Physicochemical characterization

The chemical structures of 8-arm PEG, and PEG-8SH were studied by Nuclear magnetic resonance spectrometer (NMR, AV400, Bruker, Switzerland). The dehydrated MXene, MXene/PEDOT:PSS, MXene/PEDOT:PSS/TA films were analyzed by laser confocal Raman spectrometer (Raman, LabRAM HR800, Horiba Jobin Yvon, France) under 532 nm laser. The dehydrated MXene, MXene/PEDOT:PSS, MXene/PEDOT:PSS/TA films were also analyzed by X-ray photoelectron spectroscopy (XPS, AXIS-ULTRA DLD, Kratos, Japan). The peaks of elements C and Ti were analyzed semi-quantitatively by XPSpeak 41. The dehydrated MXene, MXene/PEDOT:PSS, MXene/PEDOT:PSS/TA films were analyzed by atomic force microscopic (AFM, Jupiter XR, Asylum Research, US) to investigate height and current.

### Rheological measurements

Rheological experiments were conducted using a rheometer (MCR102, Anton Paar, Austria) set to operate at 37 °C. In brief, 200 μL of the ICAA precursor was applied onto the rheometer. Time sweep measurements were conducted with a constant amplitude of 10% at 1 Hz. The gelation time was defined as the point at which G′ (storage modulus) equaled G″ (loss modulus). And the influence of environmental variables on gelation time of ICAA hydrogel, including temperature, pH levels, ionic strength, were also investigated. Angular frequency sweep measurements were conducted over a range from 0.5 to 100 rad s$^{-1}$ with a 5% amplitude. And the Young's modulus (*E*) was calculated using Eq. (1):

$$E = 2\sqrt{G'^2 + G''^2} \cdot (1 + \upsilon)$$ (1)

where *G′* and *G″* represent the storage and loss moduli at 1 Hz, respectively, and *υ*, representing Poisson's ratio, is assumed to be 0.5.

Moreover, the mechanical durability of ICAA hydrogel immersed in 1× PBS buffer (pH = 7.4 or pH = 6, at 37 °C) over 4 weeks was investigated using angular frequency sweep measurements. The measurements ranged from 0.5 to 100 rad s$^{-1}$ with a 5% amplitude.

### Mechanical characterization

Mechanical performances of ICAA hydrogels were measured on a universal testing machine (CTM8000, Xie Qiang Instrument Manufacturing, China). And the ICAA hydrogels swelled equilibrium in 1× PBS before testing. To measure the tensile properties of ICAA hydrogels, rectangular samples were prepared with length of 25 mm, width of 5 mm, and thickness of 1.5 mm. And the stretching rate was 20 mm min$^{-1}$. To measure the compressive properties of ICAA hydrogels, the cylindrical samples were prepared with a diameter of 9 mm and a height of 10 mm. The compressing rate was 50 mm min$^{-1}$ and the final strain was 80%. The elastic moduli of ICAA hydrogels were calculated from the linear section's slope of the stress-strain curve.

Adhesion properties of ICAA hydrogels on nerve tissue and PDA-coated silicone film were evaluated using a universal testing machine. To measure the adhesive strength of ICAA hydrogels, nerve tissues were sectioned into 18 × 1.7 mm rectangles, and PDA-coated silicone films were sectioned into 20 × 5 mm rectangles using a razor blade. Subsequently, these samples were adhered to polyimide (PI) films (50 × 10 mm) using cyanoacrylate-based adhesive. ICAA precursor was injected to the sample surface to initiate hydrogel gelation, followed by immediate pressing of the other same sample onto it. The overlapping area was measured 18 × 1.7 mm, and samples were cured for 10 min at 37 °C in a humid environment. Samples were then split with a rate of 20 mm min$^{-1}$. The adhesion strength of ICAA hydrogels was determined by calculating the ratio of maximum shear stress to the overlapping area between the hydrogels and the samples. To measure the interfacial toughness of ICAA hydrogels, nerve tissues and PDA-coated silicone films were cut into 18 × 1.7 mm and 20 × 5 mm rectangular sections, respectively. Subsequently, one side of either the tissue or PDA-coated silicone film was adhered to a PI film (50 × 20 mm) using cyanoacrylate-based adhesive. ICAA precursor was injected to the sample surface to initiate hydrogel gelation. Subsequently, a PI film was attached to the ICAA hydrogel using a cyanoacrylate-based adhesive, and then pulled upwards with a rate of 20 mm min$^{-1}$. The interfacial toughness of ICAA hydrogels was determined by calculating the ratio of peeling force to half of the sample width.

### Electrochemical measurement

The electrical conductivity of ICAA hydrogels was determined by four-point probe instrument (KDY-1, Kunde Semiconductor Co., Ltd.). Prior to measurements, the hydrogel samples were fully swollen in 1× PBS, and their conductivity was calculated using an Eq. (2):

$$\sigma = \frac{L \times I}{W \times T \times V}$$ (2)

where *I* represents the current, *V* represents the voltage, *L*, *W*, and *T* represent the length, the width, and the thickness of ICAA hydrogel, respectively.

Three-electrode system, including a working electrode (ICAA gelled on a 3 mm diameter Au electrode), a counter electrode (1 × 1 cm Pt sheet), and an Ag/AgCl reference electrode, was utilized to characterized the electrochemical properties of ICAA hydrogels in 1× PBS by an electrochemical workstation (CHI660E, Shanghai Chenhua Apparatus Shanghai Chenhua Co., Ltd.). In electrochemical impedance spectroscopy, the frequency range measured spanned from $10^{-1}$ to $10^5$ Hz, using a 5-mV amplitude sine wave current and a direct current potential of 0 V. And charge injection measurements were conducted with biphasic pulses of 1.5 ms and ±0.5 V for 100,000 cycles in 1× PBS. Charge injection density ($Q_{inj}$) was calculated using Eq. (3), based on the measured charge injection curves:

$$Q_{inj} = \frac{Q_c + Q_a}{A}$$ (3)

where $Q_c$ represents the charge amount through the cathode, $Q_a$ represents the charge amount through the anode, $A$ represents ICAA hydrogel's area, respectively. And the conductivities of ICAA hydrogel before and after cycling charge injection were measured by the four-point probe instrument

To assess electrochemical stability, ICAA hydrogels were immersed in 1× PBS (37 °C) for 4 weeks. Conductivity, charge injection, and impedance properties of the ICAA hydrogels were evaluated biweekly.

## Swelling characterization
Swelling behaviors of ICAA hydrogels were analyzed through weight measurements. The prepared ICAA hydrogels were immersed in 1× PBS for 96 h and weighed at predetermined intervals. Swelling rate (SR) was calculated using the Eq. (4):

$$SR = \frac{W_t - W_0}{W_0} \qquad (4)$$

where $W_t$ and $W_o$ are defined as the swollen weight and original weight of the hydrogel samples, respectively.

Once fully swollen, the prepared ICAA hydrogels were immersed in 1× PBS for 4 weeks to assess their stability. Mass loss (ML) was calculated using the Eq. (5):

$$ML = \frac{W_f - W_s}{W_s} \qquad (5)$$

where $W_f$ and $W_s$ are defined as the final weight and original weight of the fully swollen hydrogel samples, respectively.

## Cytocompatibility
The cytocompatibility of ICAA was detected by the CCK-8 and live-dead cell staining using PC12 pheochromocytoma derived cell line (catalog number: BNCC100235, Bena Culture Collection, China). The ICAA hydrogels were macerated in Dulbecco's modified eagle medium (DMEM) at 37 °C for 24 h to obtain the hydrogel extract (100 mg ICAA in 1 mL DMEM). The hydrogel extract was supplemented with 10% v/v fetal bovine serum and 100 U mL$^{-1}$ penicillin–streptomycin before use. PC12 cells were seeded in a 96-well tissue culture plate at a density of $5 \times 10^3$ per well and incubated at 37 °C, 5% $CO_2$ for 1 d, 3 d and 5 d at 37 °C. The cell viability was quantitatively determined by CCK-8 assay using a multi-function microplate reader (PARK 10 M, TECAN, Switzerland). In addition, cells were seeded in a 24-well tissue culture plate at a density of $2 \times 10^4$ per well and incubated at 37 °C, 5% $CO_2$ for 1 d, 3 d, and 5 d at 37 °C. The cell viability was visually assessed by live-dead staining and was investigated by fluorescence microscope image analysis.

## In vivo biocompatibility of ICAA-C neural interface
The right cervical vagus nerve of anesthetized male Sprague Dawley (SD) rats (~200 g, 7-week-old) was exposed via dissection for electrodes implantation, and the cuff electrode leads were threaded subcutaneously from the rats' backs. In the ICAA-C group ($n = 5$), ICAA-C was constructed by injecting ICAA hydrogel to fill the gap between the commercial cuff electrodes (Kedou (Suzhou) Brain-computer Technology Co., Ltd, China; inner diameter of 500 μm) and vagus nerve (diameter of ~300 μm). As a positive control ($n = 5$), commercial cuff electrodes (Kedou (Suzhou) Brain-computer Technology Co., Ltd, China; inner diameter of 300 μm), close to the diameter of the rat vagus nerve, were implanted and fixed by sutures to prevent displacement. The sham group ($n = 5$) underwent the same surgery but not implantation of electrodes.

## Construction of Myocardial infarction model
SD rats (~200 g, 7-week-old) were induced with myocardial infarction. These rats were anesthetized with pentobarbital sodium, intubated, and ventilated using an ALC-V8S ventilator (ALCBIO, China). Subsequently, their hearts were exposed via intercostal thoracotomy, with the left anterior descending artery ligated using 4.0 sutures, and echocardiography was employed to confirm the success of the MI model. The MI model rats were divided into the MI group (without therapy) ($n = 7$) and the ICAA-C group ($n = 7$) randomly. Healthy SD rats ($n = 7$) served as the control group.

## ICAA-C implantation for long-term vagus stimulation
In the ICAA-C group, the right cervical vagus nerve of anesthetized rats was exposed via dissection for electrodes implantation, and the cuff electrode leads were threaded subcutaneously from the rats' backs. Biphasic, charge-balanced rectangular current pulses (10 Hz, 0.1 mA, 1.5 ms) were applied to stimulate the vagus nerve using a biological experiment system (BL420N, Chengdu Techman) for 1 h daily over 4 weeks. During the stimulation, the rats were anesthetized with isoflurane using animal research anesthesia machine (R583S, RWD). Body surface ECG of rats was recorded throughout the procedure with subcutaneous needle electrodes connected to the BL420N system.

## Evaluation of the cardiac function and biomarkers of MI
Echocardiography for each group was conducted at various intervals, measuring LVIDd, LVIDs, EF, and FS using the machine's software. Venous blood samples were collected from the jugular vein into ice-chilled sterile centrifuge tubes on days 1, 14, and 28 post-surgeries. Serum was collected by centrifugation at 1112$g$ for 20 min at 4 °C and subsequently cryopreservation at −20 °C until analysis. Serum concentrations of CK-MB, N-terminal pro-brain natriuretic peptide (NT-proBNP), renin, angiotensin II (AngII), aldosterone (ALD), TNF-α, IL-6, noradrenaline (NE), and epinephrine (EPI) were determined using ELISA kits (Bioswamp, China).

## Histopathological and immunohistochemical examination
After 4-week implantation, cervical vagus nerves of the rats in ICAA-C, commercial cuff electrodes (inner diameter of 300 μm), and sham group were meticulously dissected from surrounding muscle tissues, and the implanted device or hydrogel were carefully removed. The obtained nerve tissues underwent fixation with 4% paraformaldehyde, dehydration, and paraffin embedding. Then the nerve segments encapsulated by the implanted device were sectioned and stained for HE histopathological analysis. S-100 (GB11359-100, Servicebio, dilution 1:200), NFM (GB12763-100, Servicebio, dilution 1:200), and Iba-1 (GB113502, Servicebio, dilution 1:500) were selected as markers for immunofluorescence staining, assessing cell viability and inflammatory responses. DAPI was used to stain cell nuclei. ImageJ software was employed for analyzing biomarker fluorescence intensity.

After 4-week therapy, hearts, livers, spleens, lungs, and kidneys of the rats in ICAA-C, MI, and control groups underwent fixation with 4% paraformaldehyde, dehydration, and paraffin embedding. These tissues were subsequently sectioned and stained with for HE analysis. Myocardial morphology and fibrosis assessment of hearts employed Masson trichrome staining. The immunofluorescence staining of α-Actin, Cx43 (GB12234-100, Servicebio, dilution 1:300), and WGA dye were employed to evaluate myocardial repair. CAP-43 (GB11095-100, Servicebio, dilution 1:300), NF200 (GB12144-100, Servicebio, dilution 1:300), and TH (GB11181-100, Servicebio, dilution 1:200) were chosen as neuro-specific markers to assess cardiac sympathetic nerve activity, with DAPI staining the cell nuclei. Collagen I (GB11022-3-100, Servicebio, dilution 1:500), Collagen III (GB111629-100, Servicebio, dilution 1:4), TGF-β1 (GB11179-100, Servicebio, dilution 1:200), and MMP-9 (GB11132-100, Servicebio, dilution 1:500) were chosen as biomarkers to evaluate cardiac fibrosis and cell apoptosis through immunohistochemical

analysis. ImageJ software was employed for analyzing LV wall thickness, infarct area, myocardial size, immunohistochemical positivity, and biomarker fluorescence intensity.

## Statistical analysis

All experiments were performed at least three times ($n \geq 3$) for each sample. The statistical analysis was conducted with Origin 2023 software by one-way ANOVA first, and then by the Tukey' post hoc test. Data are presented as the mean ± standard deviation. The significance threshold was presented as * for $p \leq 0.05$, ** for $p \leq 0.01$, and *** for $p \leq 0.001$, respectively. NS not significant.

## Reporting summary

Further information on research design is available in the Nature Portfolio Reporting Summary linked to this article.

## Data availability

All data supporting the findings of this study are available within the paper and its supplementary information or from the corresponding authors on request. Source data are provided with this paper.

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

## Acknowledgements

This work was supported by National Natural Science Foundation of China (82151316, L. Z. and 81771974, Z. L.).

## Author contributions

M. Y., L. Z., Y. J., and Z. L. designed and conceptualized the study. M. Y., L. W., Q. J., L. Z., Y. J., and Z. L. supervised the overall conception and design. M. Y. and Y. H. contributed visualization. M. Y., Y. J., and Z. L. wrote and revised the paper. All authors commented on the paper.

## Competing interests

The authors declare no competing interests.
