## [Peer Review File · Nature Communications]

Reviewers' Comments:

Reviewer #1:

Remarks to the Author:

The work by Yang et al. presents a stable, injectable, conductive hydrogel bioelectronics for chronic neuromodulation. The authors utilized a multifunctional molecular regulator to enhance the mechanical stability and conductivity of the hydrogel, demonstrating its efficacy through in vivo experiments on a rat myocardial infarction model. These findings could potentially contribute to the field of bioelectronic medicine by offering a less invasive alternative for neuromodulation.

Below are some detailed comments and suggestions.

1. While the development of injectable conductive hydrogels is not new (DOI: 10.1126/science.adc9998, doi.org/10.1002/adma.201904752), the authors should emphasize the aspects of the hydrogel composition and its specific application to neuromodulation. Detailing the unique properties of the tannic acid-modulated reaction kinetics and their effects on the hydrogel's functionality compared to existing materials will help establish the novelty. It would be beneficial to include a comparative analysis or a table summarizing how this hydrogel improves upon or differs from those reported in recent studies.
2. Given that the physiological environment within a biological system is complex and variable, encompassing factors such as temperature, pH levels, ionic strength, and the presence of various biological molecules, how might these factors influence the gelation time of the hydrogel? The manuscript should also include the impact of these environmental variables on gelation time, rather than focusing solely on the effect of cross-linking density.
3. Mechanical mismatches could lead to stress concentration between the implant and surrounding tissues, affecting long-term biocompatibility and functionality. The manuscript should explore in greater detail how the mechanical properties of the hydrogel (e.g., compressive modulus, tensile strength, elongation at break) match those of the target biological tissues (e.g., neural, cardiac tissues, especially nerve fibers).
4. The authors should evaluate the mechanical behavior of the hydrogel under simulated physiological conditions (such as body temperature and dynamic loading) and its mechanical stability over long-term applications.
5. The authors are suggested to conduct a thorough evaluation of the hydrogel's electrochemical properties under prolonged electrical stimulation, including changes in conductivity over time and potential electrochemical degradation.
6. Although the hydrogel's excellent injectability is noted, more data are required to demonstrate its ability to maintain shape and function at the target site after injection. Shape stability in a dynamic biological environment is crucial for the durability of its

functions. Study the impact of different injection parameters (such as rate, pressure, volume) on the final form and performance of the hydrogel to optimize the injection process.

Reviewer #2:

Remarks to the Author:

General overview:

This study provides an innovative solution to contact issues and movement of nerve cuffs during chronic implantation with conductive hydrogel to bridge the gap and provide a flexible interface between the nerve and ill-fitting nerve cuff electrodes for chronic implantation. This would also be of assistance in cases on any nerve (even larger) where there is a slight variation in the size of the nerve in an animal/patient compared to the nerve cuff available. However, the hydrogel contains PEDOT and so its uses beyond animal studies are currently limited, which in turn limits its relevance if it cannot be translated to clinical studies.

It does seem this has been done before (ref 28-32) but on nerves of approx. 2mm. The main novelty of this work is that it is for much smaller nerves/nerve branches, and the hydrogel has improved conductivity and durability without compromising injectability – which is to be commended. However, the rationale of the need for this, for smaller nerves and for chronic implantation in animals only, needs to be provided and improved to clarify for readers, especially considering it cannot be used in humans.

I would also like to see the author's in-depth discussion on how this could be applied in the cases of multiple electrode arrays (i.e. how much shunting is expected for typical sizes/deviations, is it suitable at all etc.)

Despite this, it is a very in-depth study with exhaustive testing and confirmatory results. In addition, the authors performed a comprehensive study on an MI model to confirm the abilities of the hydrogel-cuff-nerve interface in chronic neuromodulation by showing improvements in cardiac outcomes.

I recommend major revision.

Major comments:

- Provide more context and rationale for needing to stimulate closer to the organ on the so-called 'fine nerves' – this is more invasive of course, and the ultimate goal would be to do this in humans? Or provide rationale for doing chronic neuromod studies in small animals specifically, and not larger (or provide evidence of small nerve branches in

larger animals, etc.). For what purpose? Why closer to organs? Lack of specificity in main trunks, etc.? The addition of PEDOT means it cannot be used in humans, and so the reasoning for requiring this in animals needs to be made clear (beyond just stating it is for chronic implantation).

- Also, clarify what you mean by “fine” – smaller than what diameter?
- Clarify in the text “fine nerves” vs “fine branches” vs “fine fibres” – used interchangeably it seems. Ensure it is correctly referred to throughout. E.g. in the abstract – “However, the small sizes and the delicate nature of nerve fibers present longstanding challenges in simplifying the fixation procedure and stabilizing the electrical-coupling interface for existing neural electrodes.” And the next line, etc. – you are not putting the electrode cuff around the fibres, but rather a certain branch (closer to the organ) of a peripheral nerve or a peripheral nerve itself. Fibres are within the fascicles within the nerve branch/nerve.
- Novel aspect – for very small nerves and “proposed a new strategy where a molecular regulator could be introduced during the formation of hydrogel networks to fine-tune the reaction kinetics. In particular, based on a biocompatible click chemistry reaction between thiol and maleimide, we discovered that tannic acid (TA) with abundant phenol groups can realize competitive binding with thiols and effectively modulate the gelation kinetics for optimal injectability while retaining excellent mechanical stability”. I think this needs to be relayed in the abstract and conclusions as it currently reads that this paper is highly novel for creating an injectable, conductive hydrogel, whereas it is mentioned that this has been done before. This study improves on what has been done (improving conductance and durability without compromising injectability) which is great, but needs to be clear that this is the main finding/novelty of the work. And, as above, clarify why it was necessary to improve this for smaller nerves/branches and only in animals – i.e. for what purpose? What do you need to do with smaller nerves and why? Why not larger nerves?
- ICAA-C – the hydrogel and cuff together – what cuff is used for this? And how precisely was this done – add to the VNS implantation/VNS section of the methods. What cuff was used when comparing sham, ICAA-C and cuff groups (commercial cuff) – is this a different cuff than that combined with the ICAA gel to form ICAA-C?

Minor comments:

Introduction:

- Intro sentence 2: as above, clarify what is considered ‘fine’ and give examples of the ‘peripheral nerve fibers adjacent to the target organ’, and as above, decide if referring to nerves or nerve branches or explain how you’d target fibres (that are within nerves) – but I believe you are meaning nerves or nerve branches, not fibres. Fix this throughout.
- E.g. of referral to fibres when it should be nerves/branches: “However, previous works have been limited to larger tissues such as rat sciatic nerves (~2 mm)²⁸⁻³⁰ and heart surfaces^{31,32}. Hydrogel bioelectronics for fine nerve fibers (<300 μm) remains elusive.”,

“We envision a combination of bio-adhesive conducting hydrogels and cuff electrodes can construct robust neural interfaces for fine nerve fibers.”, “In this approach, cuff electrodes, whose diameters are much larger than nerve fibers, can be loosely sutured to avoid stress compression, while bio-adhesive hydrogels can subsequently fill the gap between the cuff electrode and the nerve.” – it continues throughout paper.

- Intro paragraph 2: now clarify what you consider ‘thick nerves’ – above what diameter, and perhaps examples?
- So, bio-adhesive conductive hydrogels have been investigated before, but as mentioned, for 2mm and above. How many cases (which animals, humans, and types of nerves/nerve branches) would you have for requiring this to improve chronic neuromod on those smaller than 2mm (or smaller than 300um as mentioned)?
- “To this end, the hydrogel needs to be (1) injectable to fit fine nerve fibers of any shapes and sizes with reduced operational complexity;” – needs to be rewritten. The hydrogel itself is not ‘fitting’ the nerves, it is allowing for the nerve cuff to fit the nerve, despite the difference in size, so the hydrogel needs to be injectable to interface/bridge the gap/be form-fitting so that electrode cuffs can fit the nerve. This sentence confuses the reader and is not in line with the rest of the paper/point of the hydrogel and what it is doing, etc.
- This may be the download, but Fig.1 is blurry. In general, it is not clear (i.e. definition/resolution). A specific example, however, that may not just be due to the resolution, is the gap between electrode and nerve – before being filled with injectable hydrogel – this is not obvious. The two photos on the bottom left are also really small and not clear.
- Fig.1 i, ii and iii, iv and ‘left’ are referred to in text, but the figure does not contain these labels. Additionally, the abbreviations used in the text (ICAA and ICAA-C) are not used in the figure. It would aid readers if this was used in addition to the labels provided in the figure, to easily correlate the figure description and figures.
- A general reader will not understand why you are chronically stimulating the cardiac branches post-MI, and why you want to inhibit sympathetic activity, etc. A rationale/explanation/background of this should be provided in the introduction.
- “we showed that chronic neuromodulation using the ICCA-C neural interface” – ICAA-C not ICCA-C
- To do with this point – are you stimulating the vagus nerve post-MI or the cardiac branches (which makes sense with the intro above, specifying targeting nerve branches closer to the organs), clarify this. It seems it’s the cervical vagus, but this then does not correlate with the background and intro that states the ‘nerves closer to organs’. It may be for the size of the rat cervical nerve as an example – but this needs to be made clear.
- “Taken together” – the nerve cuff + hydrogel?

Results:

- Limitations of the frequency range? 1-10Hz – is this sufficient for neuromodulation

parameters for different purposes/treatments?

- The hydrogel swelling reached equilibrium in 24 hours – did the animal’s incision have to be open for this period of time to investigate? In humans or larger animals, for example, would this be feasible? Would it be safe to inject the hydrogel between the nerve and electrode cuff and close up the incision of the animal/patient?
- “Notably, the TA-enhanced interaction between the PEG network and MXene/PEDOT:PSS network further reduced the swelling ratio of the ICAA hydrogel (Supplementary Fig. 6c).” – further reduced – by how much/to what?
- “The ICAA hydrogel exhibited negligible mass loss and volume change even after a 4-week immersion in PBS at 37 °C, demonstrating its strong stability (Fig. 2g and Supplementary Fig. 7).” – is 4 weeks the max tested time? Is this how long it is stable for? How long would chronic neuromodulation be able to go on for with the hydrogel in place? Would it be safe for larger animals and humans? Note this in the results/discussion/limitations accordingly.
- Have not clarified what ICAA-1 and ICAA-2 hydrogels vs ICAA hydrogels are in text. I see it in the one-figure description but considering their names are mentioned in the text, this should be briefly clarified.
- Fig. 3o – 7 refs listed but only 3 colours visible. Are the rest just not on the scale? Or ensure you can see all, perhaps using dotted lines to overlay without the colours merging.
- Fig. 4 – the dotted blocks on the x45 images are not well-aligned with the x175 images
- “Conversely, the positive control group displayed a significant reduction in myelin-forming Schwann cells, which could be attributed to the large mechanical and geometric mismatches between rigid electrodes and soft nerve tissues.” - If changes in nerve structure/composition from commercial cuff – how is this possible if there isn’t necessarily contact with the nerve (hence rationale for hydrogel to interface)? If it was due to the stimulation, it would be consistent with the ICAA-C group. Based on the reasoning provided, how do the authors suggested the “mechanical and geometric mismatches between rigid electrodes and soft nerve tissues” cause this damage/changes in nerve? From the animal movement over time, the cuff damaging the nerve?
- Supp figure 16 – n=6 – 2 per group? Can you point out or add markers to the figures to show the fibrosis that is mentioned in text: “Similarly, the positive control group exhibited the highest degree of fibrosis, as shown by H&E staining (Supplementary Fig. 16).”
- Supp figure 17 – no difference in HR? The goal was to decrease sympathoexcitation and prevent cardiac remodelling? Was there no change in HR from activating parasympathetic vagus? Did you expect this as it is in an alive animal? I’d suggest commenting on this/making it clear. Referred to in main text “ICAA-C-based neuromodulation for myocardial rehabilitation. Chronic neuromodulation ability of ICAA-C was verified with vagus nerve stimulation in a rat model for the rehabilitation

therapy after myocardial infarction. Weight and heart rate of the rats were monitored throughout the therapy, confirming that ICAA-C- based vagus nerve stimulation showed no side effect (Supplementary Fig. 17).”

- Were any off-target effects measured or encountered? The whole vagus from cervical level was used to treat the heart.
- Supp figure 18 – referred to in text – “After the 4-week rehabilitation therapy, cardiac morphological staining was employed to assess ICAA-C's therapeutic effects on myocardial repair. H&E staining revealed mild inflammation and immune responses in the ICAA-C group when compared to the MI group (Supplementary Fig. 18).” – what changes? Can you point out in histology figures?
- Supp figure 19 – can you point out to the reader in the figures where to look to show/correlate with the wording of the results in text “Rats in MI group showed significant ventricular dilatation, evidenced by an increase in left ventricular internal diameter and volume at both end-diastole (LVIDd and EDV) and end-systole (LVIDs and ESV).” – i.e. showing the area where ventricular dilation is measured, etc. Is this perhaps the dotted lines shown in Fig 5a? If so, explain in figure description, and add to suppl figures in the same way (dotted lines in figure and figure description).
- “These two biomarkers were significantly higher in the MI group compared to the control group (Fig.5f, g). Concurrently, renin, angiotensin II (Ang-II), and aldosterone (ALD) related to heart failure, elevated in the MI group compared to the control group 4 weeks post-MI (Fig.5h-j). Vagus nerve stimulation enabled by ICAA- C markedly led to the downregulation of these markers, further indicating its therapeutic effect.” – by what % was this changed post VNS?
- Spelling errors in Suppl. Info
- Can you remove the nerve cuff after time, when this hydrogel has been used?

Methods:

- MI model? Further explanation for reproducibility
- In general in methods – it needs to be clarified which nerves/part of nerves were dissected and stained (e.g. cervical? At level of cuff placement? Whole nerve dissected but where and at what intervals, how close to device, etc.), which veins (for venous blood for ELISA?) Was the device + hydrogel removed? Are there histology sections from midway in the cuff?
- Rationale of study is to stimulate close to organs, hence small nerve branches – hence the need for hydrogel between nerve and nerve cuff. But the cervical vagus nerve is used? If the explanation is that the cervical vagus of rats is a size similar to ‘fine’ branches/nerves that are closer to the organs in larger animals – this needs to be clarified. It is unclear if the goal is small/fine nerves closer to organs in larger animals and humans, or if its for small/fine nerves closer to organs in small animals, but then why use cervical vagus nerve? Takes it back to first major comment. It is highly confusing at the moment.

- Unless a requirement by the journal in its current format, I would suggest combining all methods together, in order, at the end of the manuscript or in the supplementary info. At the moment, it is split, and it is confusing.

Reviewer #3:

Remarks to the Author:

This manuscript is an interesting piece of work. The research of “Highly-stable, injectable, conductive hydrogel bioelectronics for chronic neuromodulation” is reported in detail. Several comments are shown as follows:

1. Supplementing the data statistical analysis software in the article is suggested.
2. In Fig. 5k, the middle area in the MI group exhibited obvious red fluorescence. Was this due to non-specific staining? The authors should explain this or change it. The general consensus in the field is that the CX43 protein should be distributed in a star pattern.
3. The author mentioned, “The ICAA hydrogel exhibited negligible mass loss and volume change even after a 4-week immersion in PBS at 37 °C, demonstrating its strong stability”. It has been reported that CMs are seriously hypoxic after MI and aerobic respiration generates a large amount of lactic acid at the same time, resulting in an acidic microenvironment (pH < 6.8) in the infarcted area. Whether the ICAA hydrogel remained stable under an acidic microenvironment after MI, the author may give a more detailed comment or discussion on the results of stability.
4. The claim that “ICAA hydrogels possess dense cross-linking that restricts the diffusion of polymers and feature a smooth wet surface. This property helps to avoid unwanted adhesion to surrounding tissues and reduces the likelihood of post-operative adhesions”. It is suggested to add the manifestation of asymmetric or janus adhesion.
5. The authors argue that the ICAA hydrogel shows anti-swelling and long-term stability to ensure mechanical and electrical durability. It is recommended to supplement the experiments related to mechanical durability.
6. “Conversely, ICAA-C-based vagus nerve stimulation reduced LVIDd and LVIDs, mitigating ventricular remodeling through mechanical support”. Although the effect of the mechanical properties of the material on ventricular remodeling cannot be excluded, the above experiments did not specifically analyze and compare the effect of mechanical support and nerve stimulation on ventricular remodeling. Please re-elaborate and discuss.
7. “Cardiomyocytes with limited regenerative capacity compensate for the loss in number by increasing in volume in MI group, while the myocardial size in rats from the ICAA-C group exhibited a reduction, decreasing it from $689.22 \pm 105.6 \mu\text{m}^2$ to $362.74 \pm 59.99 \mu\text{m}^2$ ”. As for this description, I think it can be supported by specific experiments on cardiomyocyte hypertrophy, such as WGA immunofluorescence staining.

8. The conductive hydrogel that has been mentioned in the article regulates inflammatory factors through chronic neuro-regulation. However, TA itself has antioxidant and anti-inflammatory effects in addition to forming chemical bonds. Please elaborate on this point.

Point-by-point responses to the reviewer's comments:

Reviewer #1: *The work by Yang et al. presents stable, injectable, conductive hydrogel bioelectronics for chronic neuromodulation. The authors utilized a multifunctional molecular regulator to enhance the mechanical stability and conductivity of the hydrogel, demonstrating its efficacy through in vivo experiments on a rat myocardial infarction model. These findings could potentially contribute to the field of bioelectronic medicine by offering a less invasive alternative for neuromodulation. Below are some detailed comments and suggestions.*

Response: We feel thankful for the positive view of our article. According to the reviewer's suggestions, we have extensively revised our previous draft as detailed below.

Q1: *While the development of injectable conductive hydrogels is not new (DOI: 10.1126/science.adc9998, doi.org/10.1002/adma.201904752), the authors should emphasize the aspects of the hydrogel composition and its specific application to neuromodulation. Detailing the unique properties of the tannic acid-modulated reaction kinetics and their effects on the hydrogel's functionality compared to existing materials will help establish the novelty. It would be beneficial to include a comparative analysis or a table summarizing how this hydrogel improves upon or differs from those reported in recent studies.*

A1: Per the reviewer's suggestion, we rewrote some sentences in the introduction and the results section. Specifically, we emphasized the specific application of hydrogel to fine nerve modulation and related performance requirements for bio-adhesive hydrogel (such as injectability, mechanical stability, anti-swelling, and high and stable conductivity) on Page 4 in the revised manuscript. We also emphasized the hydrogel's composition and the role of tannic acid in improving the

hydrogel's functionalities on Pages 4-5 in the revised manuscript. More details about the regulation role of TA in hydrogel's injectability (Page 6), anti-swelling property (Page 9), adhesion (Page 10), and conductivity (Page 13-14) were also added in the revised manuscript.

Page 4: We envision a combination of bio-adhesive conducting hydrogels and cuff electrodes can construct robust neural interfaces for fine nerves. In this approach, cuff electrodes, whose diameters are much larger than nerves, can be loosely sutured to avoid stress compression, while bio-adhesive hydrogels can subsequently fill the gap between the cuff electrode and the nerves. To this end, the hydrogel needs to be (1) injectable to make the cuff fit fine nerves of any shapes and sizes with reduced operational complexity; (2) mechanically stable and anti-swelling to ensure seamless tissue integration and minimal damages; (3) highly conductive to enable effective bidirectional communication for electrical recording and stimulation.

Page 4-5: Although incorporating strong irreversible covalent bonds can enhance the mechanical stability, it tends to compromise the injectability^{38, 39}.

Herein, we proposed a new strategy where a molecular regulator could be introduced during the formation of hydrogel networks to fine tune the reaction kinetics. Typically, based on a biocompatible click chemistry reaction between thiol and maleimide, we discovered that tannic acid (TA) with abundant phenol groups can realize competitive binding with thiols and effectively modulate the gelation kinetics for optimal injectability while retaining excellent mechanical stability (Fig. 1(i)). Additionally, TA can also enhance multi-scale hydrogen bonding interactions between micron-scale electrically conducting MXene and nano-scale conducting polymer PEDOT:PSS for enhanced conductivity (Fig. 1(ii)). Furthermore, TA bridges between the hydrogel matrix and the conductive network, rendering the hydrogel

excellent anti-swelling property (Fig.1(iii)). Finally, the rich crosslinking chemistries within the network offers instant bio-adhesion (Fig. 1(iv)), allowing for highly conformal and effective electrical coupling with fine nerves.

Page 6: In our ICAA hydrogels, to slow down the reaction kinetics of -SH based click chemistry under neutral pH, we introduced TA as the molecular modulator whose abundant phenol groups can competitively react with the -SH in PEG-8SH (Fig. 2a). As a result, the ICAA hydrogel can be easily injected as a homogenous mixture onto any random surfaces, where the conformal and intimate contacts remained stable even under large deformation (Supplementary Fig. 3).

Page 9: Notably, the TA-enhanced interaction between the PEG network and MXene/PEDOT:PSS network further reduced the swelling ratio of the ICAA hydrogel to $9.22 \pm 0.76\%$ (Supplementary Fig. 7c).

Page 10: In the sol state, the precursor conformally interlocked with the contacting surface by rapidly absorbing and removing interfacial water, forming multiple dynamic non-covalent interactions, such as hydrogen bonds, due to the rich phenol groups within TA.

Page 13-14: Additionally, TA can serve as a molecular regulator to introduce abundant hydrogen bonding interactions, which facilitate the π - π stacking between PEDOT⁺ chains and MXene (Fig. 3a). As a result, compared to simple mixing, the non-covalent multi-scale conductive network strategy enabled an enhanced conductivity.

Moreover, we made a comparative photograph (as Fig. 3o) and a comparative table (as Supplementary Table 4) in terms of stability, injectability, conductivity, adhesion, and anti-swelling properties to differentiate our ICAA hydrogel from recent studies. In brief, compared to previously reported injectable conducting hydrogels based on

dynamic covalent and non-covalent bond interactions, the ICAA hydrogel, featuring a stable irreversible covalent PEG network and a multi-scale non-covalent conductive network, has significantly improved mechanical, electrochemical properties, and their stability. Meanwhile, the ICAA hydrogel demonstrates strong tissue adhesion (20.9 kPa). Therefore, the decent stability and the ability to meet various performance requirements of hydrogel bioelectronics make ICAA hydrogel advantageous than reported injectable hydrogels in neural interfaces for fine nerves. The corresponding discussion has been added on Page 18 of the revised manuscript.

Page 18: Compared to previously reported injectable conducting hydrogels based on dynamic covalent and non-covalent bond interactions^{35, 60-64}, the ICAA hydrogel, featuring a stable irreversible covalent PEG network and a multi-scale non-covalent conductive network, has significantly improved electrochemical properties and tissue-like mechanical properties (Fig. 3o and Supplementary Table 4). For instance, most injectable hydrogels for bioelectronics commonly possess low conductivities (0.065-12.5 S m⁻¹), while ICAA possesses a much-increased conductivity of 92.43 ± 7.65 S m⁻¹. Notably, different from typical injectable conductive hydrogels, the ICAA hydrogel shows anti-swelling and long-term stability to ensure mechanical and electrical durability. Meanwhile, the ICAA hydrogel also demonstrates strong tissue adhesion (20.9 kPa). The decent stability and the ability to meet various performance requirements of hydrogel bioelectronics, make ICAA hydrogel highly advantageous than previous reported injectable hydrogels in neural interface for fine nerves.

In Supporting Information:

Supplementary Table 4. Comparison of the ICAA hydrogel with previously reported injectable, conductive hydrogels.

Hydrogel	Injectability	Mechanical properties	Conductivity	Wet tissue adhesion	Anti-swelling	Stability	Reference
ICAA	Irreversible covalent bonds and reversible noncovalent bonds	6.42-40.9 kPa	$92.43 \pm 7.65 \text{ S m}^{-1}$	20.9 kPa	√	Mechanical (√) Electrical (√)	This work
CAHPs	Reversible covalent bonds and reversible noncovalent bonds	10-40 kPa	$1.35 \pm 0.32 \text{ S m}^{-1}$	4.84-13.65 kPa	√	Mechanical (/) Electrical (√)	1
IT-IC	Irreversible covalent bonds and reversible noncovalent bonds	0.1889 ± 0.0283 kPa	10 S m^{-1}	×	×	Mechanical (×) Electrical (×)	2
HPAE-Py (50%)/GelIn	Reversible covalent bonds and reversible noncovalent bonds	34.7 kPa	$0.065 \pm 0.0012 \text{ S m}^{-1}$	22.2 kPa	×	Mechanical (×) Electrical (×)	3
EGC20	Irreversible covalent bonds	3-45 kPa	/	×	×	Mechanical (×) Electrical (×)	4
γ-PGA/PEDOT: PSS	Reversible noncovalent bonds	383 kPa	12.5 S m^{-1}	/	×	Mechanical (×) Electrical (×)	5
RT-PEDOT: PSS	Reversible noncovalent bonds	1 kPa	10 S m^{-1}	×	×	Mechanical (×) Electrical (×)	6

Fig. 3o Comparison between the ICAA and other injectable conductive hydrogels according to their injection, conductivity, adhesion, and anti-swelling properties^{35, 59-63}.

Q2: *Given that the physiological environment within a biological system is complex and variable, encompassing factors such as temperature, pH levels, ionic strength, and the presence of various biological molecules, how might these factors influence the gelation time of the hydrogel? The manuscript should also include the impact of these environmental variables on gelation time, rather than focusing solely on the effect of cross-linking density.*

A2: We thank the reviewer for the instructive comments. We performed additional experiments (added to Supplementary Fig. 4) to investigate the influence of environmental variables on gelation time, including temperature, pH levels, and ionic strength. It turns out that the increase in temperature and pH will shorten the gelation time due to the accelerated reaction kinetics between the sulfhydryl group and the maleimide group under higher temperatures and alkaline conditions. Moreover, an increase in ionic strength can shield the electrostatic interactions between PEDOT and PSS, induce phase separation, and promote the gelation of PEDOT: PSS, thus also greatly shortening the gelation time of ICAA hydrogel. The corresponding discussions have been added on Page 7 of the revised manuscript.

Page 7: We further investigated the influence of environmental variables on gelation time of ICAA hydrogel, including temperature, pH levels, ionic strength (Supplementary Fig. 4b-d). It turns out that the increase of temperature and pH will shorten the gelation time, due to the accelerated reaction kinetics between the sulfhydryl group and maleimide group under higher temperature and alkaline condition. Moreover, increase of ionic strength can shield the electrostatic interactions between PEDOT and PSS, induce phase separation, and promote the gelation of PEDOT: PSS, thus also greatly shorten the gelation time of ICAA hydrogel.

In Supporting Information:

Supplementary Fig. 4 Influence of environmental variables on the gelation time of ICAA hydrogel. **a** Rheological G' and G'' variation of ICAA hydrogel show adjustable gelation time by changing its solid contents ($n = 3$). At the gelation point, the storage modulus (G') exceeds the loss modulus (G''). **b** Influence of pH on the gelation time of ICAA hydrogel ($n=3$). **c** Influence of temperature on the gelation time

of ICAA hydrogel (n=3). **d** Influence of ionic strength on the gelation time of ICAA hydrogel (n=3). The increase of pH, temperature or ionic strength shortens the gelation time of ICAA hydrogel. Data are presented as the mean \pm standard deviation in **(b)**, **(c)** and **(d)**, and were analyzed by one-way ANOVA first, and then by the Tukey's post hoc test. ***P \leq 0.001.

***Q3:** Mechanical mismatches could lead to stress concentration between the implant and surrounding tissues, affecting long-term biocompatibility and functionality. The manuscript should explore in greater detail how the mechanical properties of the hydrogel (e.g., compressive modulus, tensile strength, elongation at break) match those of the target biological tissues (e.g., neural, cardiac tissues, especially nerve fibers).*

A3: The dynamic mechanical environment of biological soft tissues involves a variety of stress types (shear, tension, and compression) and dynamic deformation (1-10 Hz, < 20%). We have tested the mechanical properties of the hydrogel by shear, tensile, and compressive tests and verified that the moduli of ICAA hydrogel on these tests are decades kPa, matching that of biological soft tissue, especially nerve tissues. Moreover, ICAA hydrogel can withstand the normal dynamic deformation process of soft tissues (1-10 Hz, < 20%). The corresponding details have been added on Pages 8-9 in the revised manuscript.

Page 8-9: Most conventional electrode materials exhibit orders of magnitude higher moduli (typically 1 MPa to 1 GPa) than that of soft physiological tissues (<100 kPa, typical elastic moduli of nerve tissues range from a few 100 Pa to about 10 kPa) and are not sufficient to provide mechanical matching interfaces^{48, 49}. Moreover, soft physiological tissues are always in a dynamic mechanical environment (strain up to 20% and frequency from 1 to 10 Hz), bearing shear, tensile and compressive forces³⁵.

Therefore, the mechanical properties of the hydrogels under shear, tensile and compressive forces were detailed investigated. The G' of the ICAA hydrogels were invariably higher than their G'' and maintain stable across the angular frequency range of 0.5 to 100 $\text{rad}\cdot\text{s}^{-1}$, indicating the ICAA hydrogel can maintain stable under dynamic conditions (Supplementary Fig. 5a). The ICAA hydrogel displayed a Young's modulus ranging from 6.42 to 40.9 kPa (Supplementary Fig. 5b). Tensile tests of ICAA hydrogel revealed an increase of ~ 1.8 times for elongation at break and change of tensile modulus from 10 ± 2 to 32.3 ± 2.9 kPa following an increase in the solid content of hydrogels (Fig. 2c, e). The same trends were also observed in compressive measurements for the strength of the ICAA hydrogel and the compressive moduli (from 42.7 to 149 kPa) when increasing the solid contents (Fig. 2d and Supplementary Fig. 6). These results indicate that the modulus and deformation of ICAA hydrogel can match that of soft biological tissues, especially nerve tissues even under dynamic physiological environment.

Q4: The authors should evaluate the mechanical behavior of the hydrogel under simulated physiological conditions (such as body temperature and dynamic loading) and its mechanical stability over long-term applications.

A4: We performed a rheological frequency sweep test for ICAA hydrogel over a frequency range from 0.5 to 100 rad s^{-1} under 37 °C, and the results were shown in Supplementary Fig. 5. The ICAA hydrogel exhibited mechanical stability under such dynamic loading and body temperature. We have supplemented additional experiments for its long-term mechanical stability; the results showed that the ICAA hydrogel possesses a stable Young's modulus during the 4-week test, indicating its long-term stability. The corresponding results have been added as Supplementary Fig.

8, and related discussions have been added on Page 10 in the revised manuscript.

Page 10: More importantly, the ICAA hydrogel exhibited a stable Young's modulus during 4-week test under dynamic loading and body temperature (pH=7.4 or pH=6), indicating its long-term mechanical durability (Supplementary Fig. 8c, d).

Supplementary Fig. 8 c Mechanical durability of ICAA hydrogel immersed in 1× PBS buffer (pH=7.4, 37 °C) during 4 weeks (n=3). **d** Mechanical durability of ICAA hydrogel immersed in 1× PBS buffer (pH=6, 37 °C) during 4 weeks (n=3).

Q5: The authors are suggested to conduct a thorough evaluation of the hydrogel's electrochemical properties under prolonged electrical stimulation, including changes in conductivity over time and potential electrochemical degradation.

A5: Since the neuromodulation process is implemented by administering a pulsed electrical stimulation, we have investigated the stability of charge injection capability (CIC) of ICAA hydrogel during prolonged biphasic pulsed electrical stimulation (10^5 cycles). The ICAA hydrogel demonstrated an insignificant decrease in performance after 10^5 charging and discharging cycles (Fig. 3j and Supplementary Fig. 17a). We have measured the conductivity of ICAA hydrogel before and after prolonged electrical stimulation, which showed no obvious change during the 10^5 -cycle test. The

corresponding result has been added as Supplementary Fig. 17b in revised Supporting Information, and a related discussion has been added on Page 17 in the revised manuscript.

Page 17: The ICAA hydrogel demonstrated an insignificant changes of CIC value under 10^5 charging and discharging cycles (Fig. 3j and Supplementary Fig. 17a). Moreover, the conductivity of the ICAA hydrogel maintained $\sim 95 \text{ S m}^{-1}$, indicating its electrical stability during prolonged stimulating test (Supplementary Fig. 17b).

In Supporting Information:

Supplementary Fig. 17 Electrochemical properties under prolonged electrical stimulation. a Changes of ICAA hydrogel's CIC during 10^5 charging and discharging cycles indicate its long-term stability under prolonged electrical stimulation. b Conductivities of ICAA hydrogel before and after 10^5 charging and discharging cycles show insignificant changes. Data are presented as the mean \pm standard deviation in (a, b) and were analyzed by one-way ANOVA first, and then by the Tukey's post hoc test. *** $P \leq 0.001$. NS, not significant.

Q6: *Although the hydrogel's excellent injectability is noted, more data are required to demonstrate its ability to maintain shape and function at the target site after injection. Shape stability in a dynamic biological environment is crucial for the durability of its*

functions. Study the impact of different injection parameters (such as rate, pressure, volume) on the final form and performance of the hydrogel to optimize the injection process.

A6: We sincerely appreciate your valuable comment, and we agree that the shape and function stability of ICAA hydrogel at the target site after injection is important. We merely characterized the function stability of ICAA hydrogel after injection (change of impedance of ICAA-C which was shown in Fig. 4e) in the previous manuscript. The ICAA-C implantation showed minimal impedance changes, while the fibrosis deposition resulted in marked changes in the impedance of the positive control group (~9.5-fold increase). In the resubmitted manuscript, we have supplemented additional experiments to investigate the shape stability of ICAA hydrogel at the target site after simple needle injection. During the 4-week implantation, the ICAA hydrogel can maintain its shape, as shown in Supplementary Fig. 19. And corresponding discussions have been added on Pages 18-19 in the revised manuscript.

In Supporting Information:

Supplementary Fig. 19 Injectable process of ICAA hydrogel. **a** Image of injection procedure of ICAA hydrogel show it can be simply injected by syringe needle. **b** Shapes of ICAA hydrogel right after implantation and 4 weeks after implantation show its *in vivo* shape stability in a dynamic biological environment.

Page 19-20: The injection procedure of ICAA hydrogel show it was simply injected by syringe needle (Supplementary Fig. 19a), and its shape demonstrated minor changes right after implantation and 4 weeks post implantation, indicating its shape stability after implantation (Supplementary Fig. 19b).

Reviewer #2: *This study provides an innovative solution to contact issues and movement of nerve cuffs during chronic implantation with conductive hydrogel to bridge the gap and provide a flexible interface between the nerve and ill-fitting nerve cuff electrodes for chronic implantation. This would also be of assistance in cases on any nerve (even larger) where there is a slight variation in the size of the nerve in an animal/patient compared to the nerve cuff available. However, the hydrogel contains PEDOT and so its uses beyond animal studies are currently limited, which in turn limits its relevance if it cannot be translated to clinical studies.*

It does seem this has been done before (ref 28-32) but on nerves of approx. 2mm. The main novelty of this work is that it is for much smaller nerves/nerve branches, and the hydrogel has improved conductivity and durability without compromising injectability - which is to be commended. However, the rationale of the need for this, for smaller nerves and for chronic implantation in animals only, needs to be provided and improved to clarify for readers, especially considering it cannot be used in humans.

I would also like to see the author's in-depth discussion on how this could be applied in the cases of multiple electrode arrays (i.e. how much shunting is expected for typical sizes/deviations, is it suitable at all etc.)

Despite this, it is a very in-depth study with exhaustive testing and confirmatory results. In addition, the authors performed a comprehensive study on an MI model to confirm the abilities of the hydrogel-cuff-nerve interface in chronic neuromodulation by showing improvements in cardiac outcomes.

I recommend major revision.

Response: We deeply appreciate the positive comments on our article. In response to the insightful suggestions from the reviewer, we have thoroughly revised our previous

draft to address the issues highlighted. Below is a detailed list of the revisions we have made.

***Q1:** Provide more context and rationale for needing to stimulate closer to the organ on the so-called “fine nerves”- this is more invasive of course, and the goal would be to do this in humans? Or provide rationale for doing chronic neuromodulation studies in small animals specifically, and not larger (or provide evidence of small nerve branches in larger animals, etc.). For what purpose? Why closer to organs? Lack of specificity in main trunks, etc.? The addition of PEDOT means it cannot be used in humans, and so the reasoning for requiring this in animals needs to be made clear (beyond just stating it is for chronic implantation).*

A1: Unselective stimulation of nerves, such as the cervical vagus containing about 100,000 nerve fibers, often causes undesired side effects (cough, hoarseness, voice alteration, anxiety, headache, and so on) (Communications Biology, 2020, 3(1): 577). Several pre-clinical studies confirmed that direct modulation of near-organ nerves, such as pancreatic nerve and splenic nerves (Nature Biotechnology, 2019, 37(12): 1446-1451; PNAS, 2021, 118(20), e2025428118; Nature Biomedical Engineering, 2022, 6(11): 1298-1316;), can improve the stimulation selectivity, thereby reducing side effects. Although the goal is to use the neural interface in humans, prior to clinical applications in humans, the effects of neural interface for fine nerves should be assessed in pre-clinical rodent models. Due to these preclinical models differ greatly from humans in the size of the anatomical site, we chose the cervical vagus nerve of a rat as a fine nerve model whose size is like humans' near-organ nerves to investigate the effects of neural interface for fine nerve.

We noted that research had been conducted with PEDOT in humans (Nature Neuroscience, 2015, 18(2): 310-315; Sci. Adv. 2016;2: e1601027; Advanced

Functional Materials, 28(12), 1700232;). The three articles we have listed demonstrated advancements in neural recording technologies, particularly focusing on high-resolution electrocorticography (ECoG) and the development of PEDOT-based electrode materials. They highlighted the advantages of using PEDOT over traditional metal electrodes, emphasizing its application in humans, such as for epilepsy surgery. Therefore, we think that it is not a problem for the application of PEDOT in humans. Context and rationale for needing to stimulate near-organ nerves have been added on Page 3 of the revised manuscript.

Page 3: Since unselective neurostimulation often causes undesired side effects (cough, hoarseness, voice alteration, anxiety, and headache and so on)⁴, selective modulation of fine peripheral nerve (< 300 μm) adjacent to the target organ, like the pancreatic and splenic nerves^{5, 6}, which offers a promising approach for treating refractory autoimmune, cardiovascular, and metabolic diseases by reducing side effects⁷⁻⁹, is urgently demanded.

Q2: Also, clarify what you mean by “fine”-smaller than what diameter?

A2: The term “fine nerves” refers to the diameter of nerve tissues, a definition that was not established by us for the first time. For example, a previous paper defined “fine nerve” as nerves with a diameter ranging from 50 to 200 μm , such as branches of the sciatic nerve, splanchnic, bladder, and vagus nerves (IEEE Transactions on Biomedical Engineering, 2015, 63(3): 581-587). Referring to this definition and in combination with our actual tested vagus nerve size of rats, we defined ‘fine’ mean nerves with a diameter smaller than 300 μm . We clarified it on Page 3 of the revised manuscript.

Page 3: ...selective modulation of fine peripheral nerve (< 300 μm) adjacent to the target organ...

Q3: Clarify in the text “fine nerves” vs “fine branches” vs “fine fibres”-used interchangeably it seems. Ensure it is correctly referred to throughout. E.g. in the abstract-“However, the small sizes and the delicate nature of nerve fibers present longstanding challenges in simplifying the fixation procedure and stabilizing the electrical-coupling interface for existing neural electrodes.” And the next line, etc.-you are not putting the electrode cuff around the fibers, but rather a certain branch (closer to the organ) of a peripheral nerve or a peripheral nerve itself. Fibers are within the fascicles within the nerve branch/nerve.

A3: We thank the reviewer for the careful read. As suggested, we have unified the terms such as fine nerves, fine branches, and fine nerve fibers as fine nerves to avoid ambiguity or inappropriate expression in our revised manuscript.

Q4: Novel aspect-for very small nerves and ‘proposed a new strategy where a molecular regulator could be introduced during the formation of hydrogel networks to fine-tune the reaction kinetics. Based on a biocompatible click chemistry reaction between thiol and maleimide, we discovered that tannic acid (TA) with abundant phenol groups can realize competitive binding with thiols and effectively modulate the gelation kinetics for optimal injectability while retaining excellent mechanical stability’. I think this needs to be relayed in the abstract and conclusions as it currently reads that this paper is highly novel for creating an injectable, conductive hydrogel, whereas it is mentioned that this has been done before. This study improves on what has been done (improving conductance and durability without compromising

injectability) which is great, but needs to be clear that this is the main finding/novelty of the work. And, as above, clarify why it was necessary to improve this for smaller nerves/branches and only in animals-i.e. for what purpose? What do you need to do with smaller nerves and why? Why not larger nerves?

A4: Limited by the word limit of the abstract, we still emphasized the innovation points in the abstract and conclusion as you suggested, namely “improving conductance and durability without compromising injectability” on Page 2 and Page 28. In addition, context and rationale for needing to stimulate near-organ nerves have been added on Page 3 of the revised manuscript, as we mentioned above.

Page 2: **Meanwhile, the mechanical and electrical stability of the hydrogel is achieved without compromising its injectability.**

Page 28: **Meanwhile, it showed substantially improved mechanical and electrical durability in contrast to previously reported injectable conductive hydrogels.**

***Q5:** ICAA-C-the hydrogel and cuff together-what cuff is used for this? And how precisely was this done-adding to the VNS implantation/VNS section of the methods. What cuff was used when comparing sham, ICAA-C, and cuff groups (commercial cuff)-is this a different cuff than that combined with the ICAA gel to form ICAA-C?*

A5: The cuff electrodes used together with ICAA hydrogel were commercial cuff electrodes with an inner diameter of 500 μm . As a positive control, commercial cuff electrodes (inner diameter of 300 μm), close to the diameter of the rat vagus nerve, were implanted and fixed by sutures to prevent displacement. In the sham group, rats underwent the same surgery but without the implantation of electrodes. Corresponding details have been added on Page 20 of the revised manuscript and in the methods section (ICAA-C implantation for long-term vagus stimulation) in the

Supporting Information.

Page 20: Specifically, ICAA-C was constructed by injecting ICAA hydrogel to fill the gap between the commercial cuff electrodes (inner diameter of 500 μm) and vagus nerves (diameter of ~ 300 μm). As a positive control, commercial cuff electrodes (inner diameter of 300 μm), close to the diameter of the rat vagus nerves, were implanted and fixed by sutures to prevent displacement. The sham group underwent the same surgery but not implantation of electrodes.

In Supporting Information:

In the ICAA-C group, ICAA-C was constructed by injecting ICAA hydrogel to fill the gap between the commercial cuff electrodes (Kedou (Suzhou) Brain-computer Technology Co., Ltd, China; inner diameter of 500 μm) and vagus nerve (diameter of ~ 300 μm). As a positive control, commercial cuff electrodes (Kedou (Suzhou) Brain-computer Technology Co., Ltd, China; inner diameter of 300 μm), close to the diameter of the rat vagus nerve, were implanted and fixed by sutures to prevent displacement. The sham group underwent the same surgery but not implantation of electrodes.

Q6: Intro sentence 2: as above, clarify what is considered 'fine' and give examples of the 'peripheral nerve fibers adjacent to the target organ', and as above, decide if referring to nerves or nerve branches or explain how you'd target fibers (that are within nerves)-but I believe you are meaning nerves or nerve branches, not fibers. Fix this throughout.

A6: In the revised introduction on Page 3, we have defined a "fine nerve", namely, a nerve with a diameter of less than 300 μm . We have also listed examples of fine nerves, such as the pancreatic and splenic nerves. Moreover, we have unified the

terms fine nerves, fine branches, and fine nerve fibers as fine nerves to avoid ambiguity or inappropriate expression in our revised manuscript.

Page 3: Since unselective neurostimulation often causes undesired side effects (cough, hoarseness, voice alteration, anxiety, and headache and so on)⁴, selective modulation of fine peripheral nerve (< 300 μm) adjacent to the target organ, like the pancreatic and splenic nerves^{5, 6}, which offers a promising approach for treating refractory autoimmune, cardiovascular, and metabolic diseases by reducing side effects⁷⁻⁹, is urgently demanded.

Q7: *E.g. of referral to fibers when it should be nerves/branches: “However, previous works have been limited to larger tissues such as rat sciatic nerves ($\sim 2\text{ mm}$)²⁸⁻³⁰ and heart surfaces^{31,32}. Hydrogel bioelectronics for fine nerve fibers (<300 μm) remains elusive.”, “We envision a combination of bio-adhesive conducting hydrogels and cuff electrodes can construct robust neural interfaces for fine nerve fibers.”, “In this approach, cuff electrodes, whose diameters are much larger than nerve fibers, can be loosely sutured to avoid stress compression, while bio-adhesive hydrogels can subsequently fill the gap between the cuff electrode and the nerve.”-it continues throughout paper.*

A7: We have unified the terms fine nerves, fine branches, and fine nerve fibers as fine nerves to avoid ambiguity or inappropriate expression in our revised manuscript, as we mentioned before.

Q8: *Intro paragraph 2: now clarify what you consider ‘thick nerves’- above what diameter, and perhaps examples?*

A8: The term “fine nerves” refers to the diameter of nerve tissues, a definition that

was not established by us for the first time. For example, a previous paper defined “fine nerve” as nerves with a diameter ranging from 50 to 200 μm , such as branches of the sciatic nerve, splanchnic, bladder, and vagus nerves (IEEE Transactions on Biomedical Engineering, 2015, 63(3): 581-587). Referring to this definition and in combination with our actual tested vagus nerve size of rats, we defined the “fine” mean nerves with a diameter smaller than 300 μm . In contrast to “fine” nerves, “thick” nerves are those with a diameter of a few millimeters, such as the sciatic nerves of rats, which are approximately 2 mm in diameter.

Q9: So, bio-adhesive conductive hydrogels have been investigated before, but as mentioned, for 2mm and above. How many cases (which animals, humans, and types of nerves/nerve branches) would you have for requiring this to improve chronic neuromodulation on those smaller than 2mm (or smaller than 300um as mentioned)?

A9: There are several pre-clinical research cases for bioelectronic modulation of nerves closer to the target organs, such as ovarian, carotid sinus nerves, greater splanchnic nerves, pancreatic nerves and splenic nerves (Communications biology, 2020, 3(1): 577; Nature Biotechnology, 2019, 37(12): 1446-1451; PNAS, 118(20), e2025428118; Nature Biomedical Engineering, 2022, 6(11): 1298-1316). The studies collectively highlight the therapeutic potential of neuromodulation in treating autoimmune diseases by targeting specific nerves in mice, rats, pigs, and human models (such as splenic and pancreatic). These studies demonstrate that precision stimulation of these nerves can modulate immune responses, reduce inflammation, and inhibit disease progression, providing a balance between efficacy and safety.

Q10: *“To this end, the hydrogel needs to be (1) injectable to fit fine nerve fibers of*

any shapes and sizes with reduced operational complexity;”-needs to be rewritten. The hydrogel itself is not “fitting” the nerves, it is allowing for the nerve cuff to fit the nerve, despite the difference in size, so the hydrogel needs to be injectable to interface/bridge the gap/be form-fitting so that electrode cuffs can fit the nerve. This sentence confuses the reader and is not in line with the rest of the paper/point of the hydrogel and what it is doing, etc.

A10: We have rewritten this sentence on Page 4 of the revised manuscript.

Page 4: To this end, the hydrogel needs to be (1) injectable to make the cuff fit fine nerve of any shapes and sizes with reduced operational complexity;

Q11: *This may be the download, but Fig.1 is blurry. In general, it is not clear (i.e. definition/resolution). A specific example, however, that may not just be due to the resolution, is the gap between electrode and nerve-before being filled with injectable hydrogel-this is not obvious. The two photos on the bottom left are also small and not clear.*

A11: We have improved the resolution of the Fig. 1 for more good reading.

Q12: *Fig.1 i, ii and iii, iv and ‘left’ are referred to in text, but the figure does not contain these labels. Additionally, the abbreviations used in the text (ICAA and ICAA-C) are not used in the figure. It would aid readers if this was used in addition to the labels provided in the figure, to easily correlate the figure description and figures.*

A13: Labels such as “i”, “ii”, “iii”, “iv”, “ICAA”, and “ICAA-C” have been added in Fig. 1 to aid readers in easily correlating the figure description and figures.

Fig. 1:

Q14: A general reader will not understand why you are chronically stimulating the cardiac branches post-MI, and why you want to inhibit sympathetic activity, etc. A rationale/explanation/background of this should be provided in the introduction.

A14: Stimulating the cardiac branches is a good option, allowing more efficient neuromodulation with fewer side effects. However, dissecting the cardiac branch of the vagus nerve in a rat model is very challenging due to its extremely small size and complex anatomical structure. This makes it difficult to avoid nerve damage during the procedure. Moreover, the cervical vagus nerve of rats is more similar in size to near-organ fine nerves in humans compared to the cardiac branches of the vagus nerve. This similarity makes the cervical vagus nerve of rats a more suitable model for studying the neural interface of fine nerves in humans. The corresponding discussion was added on Page 21 of the revised manuscript, and detailed experiments were added in the method section of the revised Supporting Information. Inhibiting the sympathetic activity of cardiac can help suppress left ventricular remodeling and a corresponding discussion has been added on Page 25 of the revised manuscript.

Page 21: We use the cervical vagus nerve of rats as a proximal organ nerve model, as

its size is similar to that of the near-organ fine nerves in humans, and the chronic neuromodulation ability of ICAA-C is verified by the rehabilitation therapy after rat myocardial infarction.

Page 25: Moreover, activation of the vagus nerve has been demonstrated to inhibit sympathetic nerve activity⁷⁰. This inhibition is essential for the treatment of MI, wherein sympathetic nerves are excessively activated and reorganized into proinflammatory circuits, contributing to left ventricular (LV) remodeling.

In Supporting Information: ...the right cervical vagus nerve of anesthetized rats was exposed via dissection for ICAA-C implantation...

Q15: “we showed that chronic neuromodulation using the ICCA-C neural interface”- ICAA-C not ICCA-C.

A15: We apologize for the typo and we have corrected it in the revised manuscript

Page 5: ...we showed that chronic neuromodulation using the ICAA-C neural interface in post-myocardial infarction therapy can reduce inflammation...

Q16: To do with this point-are you stimulating the vagus nerve post-MI or the cardiac branches (which makes sense with the intro above, specifying targeting nerve branches closer to the organs), clarify this. It seems it's the cervical vagus, but this then does not correlate with the background and intro that states the “nerves closer to organs”. It may be for the size of the rat cervical nerve as an example-but this needs to be made clear.

A16: Stimulating the near-organ nerve of the rats will allow more efficient neuromodulation with fewer side effects. However, dissecting these nerves in a rat model is very challenging due to its extremely small size and complex anatomical

structure. This makes it difficult to avoid nerve damage during the procedure. Moreover, the cervical vagus nerve of rats is more similar in size to near-organ fine nerves in humans compared to the near-organ nerve of rats. This similarity makes the cervical vagus nerve of rats a more suitable model for studying the neural interface of fine nerves in humans. The corresponding discussion and detailed experiments were added on Page 21 of the revised manuscript and the method section of the revised Supporting Information.

Page 21: We use the cervical vagus nerve of rats as a proximal organ nerve model, as its size is similar to that of the near-organ fine nerve in humans, and the chronic neuromodulation ability of ICAA-C was verified by the rehabilitation therapy after myocardial infarction.

In Supporting Information: ...the right cervical vagus nerve of anesthetized rats was exposed via dissection for ICAA-C implantation...

Q17: "Taken together"-the nerve cuff + hydrogel?

A17: We are sorry for the confusion, and we have changed it to "Overall" on Page 5 and Page 27 in our revised manuscript.

Page 5: Overall, our injectable multifunctional hydrogel bioelectronics offer promising opportunities to target various challenging anatomical locations...

Page 27: Overall, chronic stable neuromodulation consistently activates the vagus nerve and enables steady regulation of inflammatory factors and sympathetic nerve activity...

Q18: Limitations of the frequency range? 1-10Hz-is this sufficient for neuromodulation parameters for different purposes/treatments?

A18: For our in vivo electrical stimulation, the biphasic charge-balanced rectangular current pulses (10 Hz, 0.1 mA, 1.5 ms) were applied to stimulate the vagus nerve using a biological experiment system. The frequency ranges for neuromodulation are typically from 1 Hz to several hundred Hz and may be adjusted based on the patient's condition and clinical judgment. The “1 to 10 Hz” on Page 8 refers to the common deformation frequency of hydrogel after implantation rather than the neuromodulation frequency.

***Q19:** The hydrogel swelling reached equilibrium in 24 hours-did the animal's incision have to be open for this period to investigate? In humans or larger animals, for example, would this be feasible? Would it be safe to inject the hydrogel between the nerve and electrode cuff and close the incision of the animal/patient?*

A19: The gelation time and the swelling equilibrium time are not the same concept. For the feasibility and safety of injectable hydrogels, the gelation time is the more relevant indicator to consider. An appropriate gelation time allows operators sufficient time to complete the injection and positioning without difficulties caused by rapid gelation. Additionally, the hydrogel can be evenly distributed at the target site after injection. The optimal gelation time also ensures that the hydrogel remains stable in the target area, maintaining its structure and function for the necessary duration. The gelation time of our ICAA hydrogel under physiological conditions can be easily adjusted by varying the solid content, ranging from 42.4 s to 252.1 s (as shown in Fig. 2b and Supplementary Fig. 4). This range is sufficient to ensure both the feasibility of the implantation procedure and post-implantation safety. The swelling equilibrium time is simply an indicator of when the injectable hydrogel reaches a stable state after gelation. Given that the swelling equilibrium process is typically time-consuming, our

primary concern is the swelling ratio at equilibrium, as it more accurately reflects the structural stability of the hydrogel after implantation. Although the ICAA hydrogel required 24 hours to reach swelling equilibrium, it exhibited an equilibrium swelling rate of only $9.22 \pm 0.76\%$, with negligible volume changes, indicating satisfactory structural stability. Therefore, achieving an appropriate gelation time and a minimal equilibrium swelling rate can help ensure that a second surgery is not necessary to guarantee the safety of the implantation.

Q20: *“Notably, the TA-enhanced interaction between the PEG network and MXene/PEDOT:PSS network further reduced the swelling ratio of the ICAA hydrogel (Supplementary Fig. 6c).”-further reduced-by how much/to what?*

A20: TA can further reduce the swelling ratio of the ICAA hydrogel to $9.22 \pm 0.76\%$, and we added the description on Page 9 in our revised manuscript.

Page 9: **Notably, the TA-enhanced interaction between the PEG network and MXene/PEDOT:PSS network further reduced the swelling ratio of the ICAA hydrogel to $9.22 \pm 0.76\%$ (Supplementary Fig. 7c).**

Q21: *“The ICAA hydrogel exhibited negligible mass loss and volume change even after a 4-week immersion in PBS at 37 °C, demonstrating its strong stability (Fig. 2g and Supplementary Fig. 7).”-is 4 weeks the max tested time? Is this how long it is stable for? How long would chronic neuromodulation be able to go on for with the hydrogel in place? Would it be safe for larger animals and humans? Note this in the results/discussion/limitations accordingly.*

A21: The “4-week” is our designed test period, not the longest period for which the hydrogel remained stable. We demonstrated that the chronic neuromodulation can be realized with the ICAA hydrogel within 4-week rehabilitation therapy for MI.

Moreover, according to the in-vivo interface stability tests (Fig. 4 in resubmitted manuscript), ICAA-C showed no significant difference in Schwann cell and axon populations in contrast to the control group, while the positive control group displayed a significant reduction in myelin-forming Schwann cells, which could be attributed to the large mechanical and geometric mismatches between rigid electrodes and soft nerve tissues. Additionally, no significant difference was observed in Iba-1 expression among the ICAA-C and sham groups. These results indicated that the hydrogel can reduce tissue damage and alleviate the inflammatory response. In addition, the ICAA-C implantation (4 weeks) showed minimal impedance changes, while the fibrosis deposition results in marked changes in impedance of the positive control group (~9.5-fold increase) during implantation, which may eventually cause the complete failure of the devices. Therefore, we think that the ICAA hydrogel can achieve stable neural modulation for far more than 4 weeks, and it is safe for larger animals and humans. Corresponding results and discussions were added on Pages 20-21 in our revised manuscript.

Page 20: ICAA-C group showed no significant difference in Schwann cell and axon populations in contrast to the control group. Conversely, the positive control group displayed a significant reduction in myelin-forming Schwann cells, which could be attributed to the large mechanical and geometric mismatches between rigid electrodes and soft nerve tissues. Additionally, no significant difference was observed in Iba-1 expression, a crucial immune response indicator, among the ICAA-C and sham groups (Fig. 4a, d). In contrast, a more severe immune response occurred in the positive control group.

Page 21: Moreover, the ICAA-C implantation showed minimal impedance changes, while the fibrosis deposition results in marked changes in impedance of the positive

control group (~9.5-fold increase), which may eventually cause the complete failure of the devices⁶⁵⁻⁶⁷.

Q22: *Have not clarified what ICAA-1 and ICAA-2 hydrogels vs ICAA hydrogels are in text. I see it in the one-figure description but considering their names are mentioned in the text, this should be briefly clarified.*

A22: ICAA-1 and ICAA-2 were clarified in text on Pages 6-7 in the resubmitted manuscript.

Page 6-7: We investigated the injectability, mechanical properties, and stability of our ICAA hydrogels in detail by preparing samples with varied solid contents of PEG-8SH, PEG-2Mal, and MXene (ICAA (15 wt%, 10 wt% and 4 wt%); ICAA-1 (5 wt%, 3.33 wt% and 1.33 wt%); ICAA-2 (10 wt%, 6.6 wt%, and 2.66 wt%)) (Supplementary Table 1).

Q23: *Fig. 3o-7 refs listed but only 3 colors visible. Are the rest just not on the scale? Or ensure you can see all, perhaps using dotted lines to overlay without the colors merging.*

A23: We thank the reviewer for the suggestion. We have changed Fig.3o into dotted lines in the revised manuscript.

Fig. 3o Comparison between the ICAA and other injectable conductive hydrogels according to their injection, conductivity, adhesion, and anti-swelling properties^{35, 59-63}.

Q24: Fig. 4-the dotted blocks on the x45 images are not well-aligned with the x175 images.

A24: We made the dotted blocks on the x45 images that well-aligned with the x175 images in Fig.4 in our revised manuscript.

Fig. 4 Stable neural interface constructed by ICAA-C. **a** Representative immunofluorescence photographs of the vagus nerve at 4 weeks post-implantation in ICAA-C, commercial cuff (diameter of 300 μm), and sham group. The images are magnified $\times 45$ (left), $\times 175$ (right) to show the white dotted regions of the nerves. Scale bar, 20 μm (left), 10 μm (right)....

Q25: “Conversely, the positive control group displayed a significant reduction in myelin-forming Schwann cells, which could be attributed to the large mechanical and geometric mismatches between rigid electrodes and soft nerve tissues.”- If changes in

nerve structure/composition from commercial cuff-how is this possible if there isn't necessarily contact with the nerve (hence rationale for hydrogel to interface)? If it was due to the stimulation, it would be consistent with the ICAA-C group. Based on the reasoning provided, how do the authors suggested the "mechanical and geometric mismatches between rigid electrodes and soft nerve tissues" cause this damage/changes in nerve? From the animal movement over time, the cuff damaging the nerve?

A25: Under normal circumstances, to construct a stable neural interface, cuff electrodes that approach the size of the target nerve are used. To simulate the actual situation, as a positive control in our study, commercial cuff electrodes (inner diameter of 300 μm), close to the diameter of the rat vagus nerve, were implanted and fixed by sutures to prevent displacement. However, nerve size has individual differences; such cuff electrodes with standard size cannot perfectly fit the nerve. Moreover, suture fixation easily causes the electrodes to exert stress compression on the nerves. The exerted stress compression due to size mismatch and suture fixation should be the reasons for nerve damage in the positive control group, which is consistent with previous research reported (Nature Communications, 2023, 14:2206).

Q26: *Supp figure 16- n=6-2 per group? Can you point out or add markers to the figures to show the fibrosis that is mentioned in text: "Similarly, the positive control group exhibited the highest degree of fibrosis, as shown by H&E staining (Supplementary Fig. 16)."*

A26: n=6 means 6 samples per group in Supplementary Fig. 20. We added red dotted boxes as markers to show the fibrosis in the nerve section images in the revised Supporting Information.

Supplementary Fig. 20... The fibrosis regions are marked with red dotted boxes...

Q27: Supp figure 17-no difference in HR? The goal was to decrease sympathy excitation and prevent cardiac remodeling? Was there no change in HR from activating parasympathetic vagus? Did you expect this as it is in an alive animal? I'd suggest commenting on this/making it clear. Referred to in main text "ICAA-C-based neuromodulation for myocardial rehabilitation. Chronic neuromodulation ability of ICAA-C was verified with vagus nerve stimulation in a rat model for the rehabilitation therapy after myocardial infarction. Weight and heart rate of the rats were monitored throughout the therapy, confirming that ICAA-C-based vagus nerve stimulation showed no side effect (Supplementary Fig. 17)."

A27: Based on the previous experimental results of our research group, high-dose stimulation of the cervical vagus nerve can induce changes in the heart rate of rats. Additionally, we observed side effects in rats, such as coughing, hoarseness, and voice alterations. In this study, we chose a low-dose stimulation parameter (100 μ A), and we did not observe a significant change in the heart rate of the rats throughout the

vagus nerve stimulation treatment. Our results confirm that vagus nerve stimulation can affect the autonomic nervous system of the heart even at low doses without affecting heart rate, which is consistent with several previous studies (International Heart Journal, 2016, 57(3): 350-355; American Journal of Physiology-Heart and Circulatory Physiology, 2015, 309(7): H1198-H1206). This is due to the vagus nerve stimulation regulating the balance between the sympathetic and parasympathetic nerves. Specifically, one of these studies investigated how vagus nerve stimulation aids in rehabilitation after myocardial infarction. The findings indicated that it improves calcium handling, reduces remodeling of the intrinsic cardiac nervous system, and decreases sympathetic nerve activity without significantly affecting heart rate (International Heart Journal, 2016, 57(3): 350-355). Another listed study explores the impact of chronic vagus nerve stimulation on the remodeling of the intrinsic cardiac nervous system and cardiac tissue post-myocardial infarction in guinea pigs without affecting heart rate. The findings demonstrated that vagus nerve stimulation prevented cardiac function and mitigated the depolarization of resting membrane potentials and the increased input resistance in intrinsic cardiac nervous induced by MI. (American Journal of Physiology-Heart and Circulatory Physiology, 2015, 309(7): H1198-H1206).

***Q28:** Were any off-target effects measured or encountered? The whole vagus from cervical level was used to treat the heart.*

A28: Stimulation of the whole vagus from the cervical level was applied to treat the heart in this study. Since we chose a low-dose stimulation parameter (100 μ A), we did not observe severe side effects.

Q29: *Supp figure 18-referred to in text- “After the 4-week rehabilitation therapy, cardiac morphological staining was employed to assess ICAA-C's therapeutic effects on myocardial repair. H&E staining revealed mild inflammation and immune responses in the ICAA-C group when compared to the MI group (Supplementary Fig. 18).” -what changes? Can you point out in histology figures?*

A29: “Inflammation and immune responses” here are preliminarily determined by the changes they cause in the structure of heart tissue, characterized by myocardial cell necrosis, loose tissue, and increased transparency. We have pointed out the necrotic and loose areas of tissue with red arrows in the tissue section in Supplementary Fig. 22, and a detailed discussion has been added on Page 21 in the revised manuscript.

Page 21: H&E staining revealed mild inflammation and immune responses in the ICAA-C group when compared to the MI group, **characterized by myocardial cell necrosis, loose tissue and increased transparency** (Supplementary Fig. 22).

In Supporting Information:

Supplementary Fig. 22 Histological analysis of hearts (n = 6). H&E-stained histological appearance of the heart tissues dissected in control group (a), MI group

(b), and ICAA-C group (c). The image was magnified $\times 1$ (top), $\times 15$ (middle and bottom) to show upper (blue) and lower (green) regions of the left ventricular wall. Scale bars, 1 mm (top), 100 μm (middle and bottom). **H&E staining revealed mild inflammation and immune responses (pointed out by red arrows), characterized by myocardial cell necrosis, loose tissue and increased transparency.**

Q30: Supp figure 19-can you point out to the reader in the figures where to look to show/correlate with the wording of the results in text “Rats in MI group showed significant ventricular dilatation, evidenced by an increase in left ventricular internal diameter and volume at both end-diastole (LVIDd and EDV) and end-systole (LVIDs and ESV).” -i.e. showing the area where ventricular dilation is measured, etc. Is this perhaps the dotted lines shown in Fig 5a? If so, explain in figure description, and add to suppl figures in the same way (dotted lines in figure and figure description).

A30: The results text here referred to echocardiography (Fig. 5a and Supplementary Fig. 23). The dotted lines showed the size changes of the left ventricle in the cross-section at the level of the papillary muscles during systole and diastole. During the test, the instrument can calculate the corresponding cardiac function parameters, including LVIDd, EDV, LVIDs, ESV and so on. The corresponding figure description has been added on Page 23 of the revised manuscript. The figure description and dotted lines have been added for Supplementary Fig. 23 in the revised Supporting Information.

Page 23: ...The dotted lines showed the size changes of the left ventricle in the cross-section at the level of the papillary muscles during systole and diastole....

In Supporting Information:

Supplementary Fig. 23 Echocardiography imaging (n = 7) of rats in the control, MI, and ICAA-C groups at day 1 (a) and day 14 (b) after modeling. The dotted lines showed the size changes of the left ventricle in the cross-section at the level of the papillary muscles during systole and diastole. Scale bar, 5 mm.

Q31: “These two biomarkers were significantly higher in the MI group compared to the control group (Fig.5f, g). Concurrently, renin, angiotensin II (Ang-II), and aldosterone (ALD) related to heart failure, elevated in the MI group compared to the control group 4 weeks post-MI (Fig.5h-j). Vagus nerve stimulation enabled by ICAA-C markedly led to the downregulation of these markers, further indicating its therapeutic effect.” -by what % was this changed post VNS?

A31: Each of these biomarkers decreased by more than 15% post VNS. The corresponding discussion has been added on Page 22 of the revised manuscript.

Page 22: Vagus nerve stimulation enabled by ICAA-C markedly led to the downregulation of these markers by more than 15%, further indicating its therapeutic effect.

Q33: Spelling errors in Suppl. Info

A33: We apologize to the typos and we have revised them in the current version.

Q34: Can you remove the nerve cuff after time, when this hydrogel has been used?

A34: The cuff electrodes can be removed with careful dissection procedures even after hydrogel has been used.

Q35: MI model? Further explanation for reproducibility.

A35: These experimental details about the construction of the MI model were added in the revised Supporting Information.

In Supporting Information: Rats were anesthetized with pentobarbital sodium, intubated, and ventilated using an ALC-V8S ventilator (ALCBIO, China). Subsequently, their hearts were exposed via intercostal thoracotomy, with the left anterior descending artery ligated using 4.0 sutures, and echocardiography was employed to confirm the success of the MI model.

Q36: In general, in methods-it needs to be clarified which nerves/part of nerves were dissected and stained (e.g. cervical? At level of cuff placement? Whole nerve dissected but where and at what intervals, how close to device, etc.), which veins (for venous blood for ELISA?) Was the device + hydrogel removed? Are there histology sections from midway in the cuff?

A36: The cervical vagus nerves were dissected, and the nerve segments encapsulated by the implanted device were used for subsequent section and staining after carefully removing the device and hydrogel. The venous blood was collected from collected from the jugular vein. The corresponding details were added in the method section in the revised Supporting Information.

In Supporting Information: After 4-week implantation, **cervical** vagus nerves of the rats in ICAA-C, commercial cuff electrodes (inner diameter of 300 μm), and sham group were meticulously dissected from surrounding muscle tissues, and **the implanted device or hydrogel were carefully removed**. The obtained tissues underwent fixation with 4% paraformaldehyde, dehydration, and paraffin embedding. Then **the nerve segments encapsulated by the implanted device** were sectioned and stained with for HE histopathological analysis.

In Supporting Information: Venous blood samples were **collected from the jugular vein** into ice-chilled sterile centrifuge tubes on days 1, 14, and 28 post-surgeries.

Q37: Rationale of study is to stimulate close to organs, hence small nerve branches-hence the need for hydrogel between nerve and nerve cuff. But the cervical vagus nerve is used? If the explanation is that the cervical vagus of rats is a size similar to 'fine' branches/nerves that are closer to the organs in larger animals-this needs to be clarified. It is unclear if the goal is small/fine nerves closer to organs in larger animals and humans, or if it's for small/fine nerves closer to organs in small animals, but then why use cervical vagus nerve? Takes it back to first major comment. It is highly confusing at the moment.

A37: We chose the cervical vagus of rats as a fine nerve model because its size is similar to fine nerves that are closer to the organs in larger animals, and we clarified it

on Page 21 of our revised manuscript.

Page 21: We use the cervical vagus nerve of rats as a proximal organ nerve model, as its size is similar to that of the near-organ fine nerve in humans, and the chronic neuromodulation ability of ICAA-C was verified by the rehabilitation therapy after myocardial infarction.

Q38: Unless a requirement by the journal in its current format, I would suggest combining all methods together, in order, at the end of the manuscript or in the supplementary info. At the moment, it is split, and it is confusing.

A38: We have combined all methods together in the revised Supporting Information according to the reviewer's suggestion.

Reviewer #3: *This manuscript is an interesting piece of work. The research of “Highly-stable, injectable, conductive hydrogel bioelectronics for chronic neuromodulation” is reported in detail. Several comments are shown as follows:*

We greatly appreciate the reviewer for his/her interest in our study and the instructive comments to improve our work. We have carefully revised the manuscript according to the comments.

Q1: *Supplementing the data statistical analysis software in the article is suggested.*

A1: The data statistical analysis software used was Origin 2023 software, and it was clarified in the method section in the revised Supporting Information.

In Supporting Information: The statistical analysis was conducted with Origin 2023 software by one-way ANOVA first, and then by the Tukey’ post hoc test.

Q2: *In Fig. 5k, the middle area in the MI group exhibited obvious red fluorescence. Was this due to non-specific staining? The authors should explain this or change it. The consensus in the field is that the CX43 protein should be distributed in a star pattern.*

A2: The obvious red fluorescence in the middle area in the MI group was due to non-specific staining, and we replaced it to avoid this problem in Fig. 5k on Page 23 in our revised manuscript.

Fig. 5k:

Q3: *The author mentioned, “The ICAA hydrogel exhibited negligible mass loss and*

volume change even after a 4-week immersion in PBS at 37 °C, demonstrating its strong stability”. It has been reported that CMs are seriously hypoxic after MI and aerobic respiration generates a large amount of lactic acid at the same time, resulting in an acidic microenvironment (pH<6.8) in the infarcted area. Whether the ICAA hydrogel remained stable under an acidic microenvironment after MI, the author may give a more detailed comment or discussion on the results of stability.

A3: We did additional experiments related to the stability of ICAA hydrogel under an acidic environment (added to Supplementary Fig. 8), and the results indicated that the ICAA hydrogel can remain stable under an acidic environment for 4 weeks. The corresponding results and discussions were added to Page 10 of our revised manuscript.

Page 10: More importantly, the ICAA hydrogel exhibited a stable Young’s modulus during 4-week test under dynamic loading and body temperature (pH=7.4 or pH=6), indicating its long-term mechanical durability (Supplementary Fig. 8c, d).

In Supporting Information:

Supplementary Fig. 8 ...d Mechanical durability of ICAA hydrogel immersed in 1X PBS buffer (pH=6, 37 °C) during 4 weeks....

Q4: The claim that “ICAA hydrogels possess dense cross-linking that restricts the

diffusion of polymers and feature a smooth wet surface. This property helps to avoid unwanted adhesion to surrounding tissues and reduces the likelihood of post-operative adhesions". It is suggested to add the manifestation of asymmetric or Janus adhesion.

A4: The images showing the Janus adhesion property of ICAA hydrogel were added as Supplementary Fig. 10 in the revised Supporting Information. The corresponding discussion has been added on Page 11 of the revised manuscript.

Page 11: In the gel state, ICAA hydrogels possess dense cross-linking that restricts the diffusion of polymers and features a smooth wet surface. This property helps to avoid unwanted adhesion to surrounding tissues and reduces the likelihood of post-operative adhesions (Supplementary Fig. 10).

In Supporting Information:

Supplementary Fig. 10 Janus adhesion of the ICAA hydrogel (after fully gelled) on diverse tissues. a Janus adhesion of the ICAA hydrogel with rat's muscles. **b** Janus adhesion of the ICAA hydrogel with rat's nerves. In the gel state, ICAA hydrogels possess dense cross-linking that restricts the diffusion of polymers and feature a smooth wet surface, avoiding unwanted adhesion to surrounding tissues and reduces the likelihood of post-operative adhesions.

Q5: *The authors argue that the ICAA hydrogel shows anti-swelling and long-term stability to ensure mechanical and electrical durability. It is recommended to supplement the experiments related to mechanical durability.*

A5: We did additional experiments related to the mechanical durability of ICAA hydrogel, and the results indicated that the ICAA hydrogel can maintain mechanical stability during 4 weeks. The corresponding results and discussions were added on Page 10 of our revised manuscript.

Page 10: More importantly, the ICAA hydrogel exhibited a stable Young's modulus during 4-week test under dynamic loading and body temperature (pH=7.4 or pH=6), indicating its long-term mechanical durability (Supplementary Fig. 8c, d).

Supplementary Fig. 8 ...c Mechanical durability of ICAA hydrogel immersed in 1X PBS buffer (pH=7.4, 37 °C) during 4 weeks....

Q6: *“Conversely, ICAA-C-based vagus nerve stimulation reduced LVIDd and LVIDs, mitigating ventricular remodeling through mechanical support”. Although the effect of the mechanical properties of the material on ventricular remodeling cannot be excluded, the above experiments did not specifically analyze and compare the effect of mechanical support and nerve stimulation on ventricular remodeling. Please re-elaborate and discuss.*

A6: ICAA hydrogel was injected into the cervical vagus nerve site instead of the MI area to act as a neural interface between cuff electrodes and the cervical vagus nerve. The term "mechanical support" here refers to the maintenance of the heart's own size. To avoid ambiguity, we have removed this term on Page 22 of the revised manuscript.

Page 22: Conversely, ICAA-C-based vagus nerve stimulation reduced LVIDd and LVIDs, mitigating ventricular remodeling (Supplementary Fig. 24, 25).

Q7: *“Cardiomyocytes with limited regenerative capacity compensate for the loss in number by increasing in volume in MI group, while the myocardial size in rats from the ICAA-C group exhibited a reduction, decreasing it from $689.22 \pm 105.6 \mu\text{m}^2$ to $362.74 \pm 59.99 \mu\text{m}^2$ ”. As for this description, I think it can be supported by specific experiments on cardiomyocyte hypertrophy, such as WGA immunofluorescence staining.*

A7: In the previous manuscript, we used actin immunofluorescence staining to evaluate the size change of cardiomyocytes. We have supplemented WGA immunofluorescence staining to further investigate the size change of cardiomyocytes more accurately. Cardiomyocytes with limited regenerative capacity compensate for the loss in number by increasing in volume in the MI group (from $382.32 \pm 45.91 \mu\text{m}^2$ to $810.35 \pm 123.43 \mu\text{m}^2$), while the myocardial size in rats from the ICAA-C group exhibited a reduction, decreasing it to $439.19 \pm 119.59 \mu\text{m}^2$. The corresponding results have been added as Fig. 5m (in revised manuscript) and as Supplementary Fig. 26 (in the revised Supporting Information). The corresponding discussion has been added on Page 24 of the revised manuscript.

Page 24: Cardiomyocytes with limited regenerative capacity compensate for the loss in number by increasing in volume in MI group, while the myocardial size in rats

from the ICAA-C group exhibited a reduction, decreasing it from $810.35 \pm 123.43 \mu\text{m}^2$ to $439.19 \pm 119.59 \mu\text{m}^2$ (Fig. 5m, and Supplementary Fig. 26).

In Supporting Information:

Supplementary Fig. 26 Representative immunofluorescence images of the infarct area in the ICAA-C, MI, and control groups for cardiac markers (WGA) and nuclei (DAPI). ICAA-C-based vagus nerve stimulation suppress their enlargement of cardiomyocytes after MI. Scale bar, 20 μm .

Fig. 5 ...l, m, n, o Quantitative analysis of Cx43 positive area% (l), myocyte size (m), transforming growth factor beta 1 (TGF- β 1) positive area% (n), and matrix metalloproteinase 9 (MMP-9) positive area% (o).

Q8: The conductive hydrogel that has been mentioned in the article regulates inflammatory factors through chronic neuro-regulation. However, TA itself has antioxidant and anti-inflammatory effects in addition to forming chemical bonds.

Please elaborate on this point.

A8: ICAA hydrogel was injected into the cervical vagus nerve site instead of the MI area to act as a neural interface between cuff electrodes and the cervical vagus nerve. More details about the implantation procedures of ICAA hydrogel were added in the method section in the revised Supporting Information. Although TA can act as an anti-oxidant and anti-inflammatory agent, we think inflammation caused by MI was mainly regulated by chronic neuromodulation as TA was not directly interfacing with the MI area.

In Supporting Information: The right cervical vagus nerve of anesthetized rats was exposed via dissection for electrodes implantation, and the cuff electrode leads were threaded subcutaneously from the rats' backs. In the ICAA-C group, ICAA-C was constructed by injecting ICAA hydrogel to fill the gap between the commercial cuff electrodes (Kedou (Suzhou) Brain-computer Technology Co., Ltd, China; inner diameter of 500 μm) and vagus nerve (diameter of ~ 300 μm). As a positive control, commercial cuff electrodes (Kedou (Suzhou) Brain-computer Technology Co., Ltd, China; inner diameter of 300 μm), close to the diameter of the rat vagus nerve, were implanted and fixed by sutures to prevent displacement. The sham group underwent the same surgery but not implantation of electrodes.

List of Changes according to the Comments

1. The specific application of hydrogel to fine nerve modulation and related performance requirements for bio-adhesive hydrogel were emphasized on Page 4 of the revised manuscript.
2. The hydrogel composition and the role of tannic acid in improving hydrogel's functionalities were added on Pages 4-5 in the revised manuscript.
3. More details about TA regulation's role in hydrogel's injectability (Page 6), anti-swelling property (Page 9), adhesion (Page 10), and conductivity (Page 13) were added in the revised manuscript.
4. A comparative photograph (as Fig. 3o) and a comparative table (as Supplementary Table 4) to differentiate our ICAA hydrogel from recent studies were added in the revised manuscript and Supporting information.
5. We supplemented experiments (added into Supplementary Fig. 4) to investigate the influence of environmental variables on gelation time, and the corresponding discussions were on Page 7 in the revised manuscript.
6. The corresponding results of ICAA hydrogel's mechanical durability were added as Supplementary Fig. 8, and related discussions were added on Page 10 in the revised manuscript.
7. The changes in ICAA hydrogel's CIC and conductivity during prolonged electrical stimulation were as Supplementary Fig. 17 in the revised Supporting Information, and related discussion was on page 17 in the revised manuscript.
8. The injection process and shape stability of ICAA hydrogel in a dynamic biological environment were added as Supplementary Fig. 19, and related discussion was on Pages 18-19 in the revised manuscript.
9. Context and rationale for needing to stimulate near-organ nerves were added on

Page 3 of the revised manuscript.

10. We defined the “fine” mean nerves with a diameter smaller than 300 μm , and we clarified it on Page 3 of the revised manuscript.
11. We unified the terms fine nerves, fine branches, and fine nerve fibers as fine nerves in the revised manuscript.
12. We emphasized the innovation point, namely “improving conductance and durability without compromising injectability”, on Page 2 and Page 28 in the revised manuscript.
13. More details about cuff electrodes used in our study were added on Page 20 of the revised manuscript and in the methods section in the revised Supporting Information.
14. We rewrote the sentence, namely, “To this end, the hydrogel needs to be (1) injectable to make the cuff fit fine nerves of any shapes and sizes with reduced operational complexity” on Page 4 of the revised manuscript.
15. We improved the resolution of Fig. 1 and added more labels for better reading.
16. The corresponding discussion about why the rat’s cervical vagus nerve was a fine nerve model was added on Page 21 of the revised manuscript, and detailed experiments were added in the method section in the revised Supporting Information.
17. The corresponding discussion about why inhibit sympathetic activity was added on Page 25 in the revised manuscript.
18. The typos on Page 5 and Page 27 were revised.
19. The specific value of swelling ratio reduction of the ICAA hydrogel by TA was added on Page 9 in the revised manuscript.
20. The corresponding results and discussions about the long-term stability and safety

- of ICAA hydrogel were added on Page 20 and Page 21 in the revised manuscript.
21. ICAA-1 and ICAA-2 were clarified in the text on Page 6 of the revised manuscript.
 22. We changed Fig. 3o into dotted lines in the revised manuscript.
 23. We made the dotted blocks on the x45 images align well with the x175 images in Fig. 4 in our revised manuscript.
 24. We added red dotted boxes as markers in Supplementary Fig. 20 to show the fibrosis in the nerve section images in revised Supporting Information.
 25. We pointed out the necrotic and loose areas of tissue with red arrows in the tissue section in Supplementary Fig. 22, and a detailed discussion has been added on Page 21 in the revised manuscript.
 26. The corresponding figure description of dotted lines has been added on Page 23 in the revised manuscript. The figure description and dotted lines have been added for Supplementary Fig. 23 in the revised Supporting Information.
 27. The specific decreasing value of biomarkers was added on Page 22 in the revised manuscript.
 28. The experimental details about the construction of the MI model, section and staining process, and blood collection process were added in the revised Supporting Information.
 29. We combined all methods together in the revised Supporting Information.
 30. The data statistical analysis software was clarified in the method section in the revised Supporting Information.
 31. We replaced the non-specific staining image in Fig. 4k in Page 23 of our revised manuscript.
 32. The images showing the Janus adhesion property of ICAA hydrogel were added as Supplementary Fig. 10 in the revised Supporting Information.

33. The term "mechanical support" was removed on Page 22 of the revised manuscript.
34. The corresponding results about WGA staining were added as Fig. 5m (in the revised manuscript) and as Supplementary Fig. 26 (in the revised Supporting Information). The corresponding discussion was added on Page 24 of the revised manuscript.
35. More details about the implantation procedures of ICAA hydrogel were added in the method section in the revised Supporting Information.

Reviewers' Comments:

Reviewer #1:

Remarks to the Author:

The authors have addressed all my comments.

The paper is suggested for publication with the following minor change:

In the title, delete bioelectronics. As it is more on the material side.

That is, "Highly-stable, injectable, conductive hydrogel for chronic neuromodulation".

Reviewer #2:

Remarks to the Author:

I recommend another revision. Most of my points were adequately addressed, however, a lot of my questions were answered in the "response to referee" but not relayed to the manuscript. I suggest the authors incorporate their answers into the manuscript itself as this reviewer did not ask those questions to satisfy their curiosity but because it was not clear to the potential reader in the manuscript itself. Specifically: Q1 (point about PEDOT applicability to humans), Q8, Q9, Q18, Q19, Q25, Q27, Q28 (with objective evidence please).

Reviewer #3:

Remarks to the Author:

This manuscript developed a highly-stable, injectable, conductive hydrogel bioelectronics for chronic neuromodulation. Given the novelty of the synergistic therapeutic strategy and the comprehensive supplementary experimental data, I would recommend accepting this manuscript.

Point-by-point responses to the reviewer's comments:

Reviewer #1: *The paper is suggested for publication with the following minor change: In the title, delete bioelectronics. As it is more on the material side. That is, "Highly-stable, injectable, conductive hydrogel for chronic neuromodulation".*

Response: We thank the reviewer for the constructive feedback on our manuscript. We have changed the title to “Highly-stable, injectable, conductive hydrogel for chronic neuromodulation”.

Reviewer #2: *I recommend another revision. Most of my points were adequately addressed, however, a lot of my questions were answered in the "response to referee" but not relayed to the manuscript. I suggest the authors incorporate their answers into the manuscript itself as this reviewer did not ask those questions to satisfy their curiosity but because it was not clear to the potential reader in the manuscript itself. Specifically: Q1 (point about PEDOT applicability to humans), Q8, Q9, Q18, Q19, Q25, Q27, Q28 (with objective evidence please).*

Response: We thank the reviewer for the comment. We have now incorporated our responses to those questions into the revised manuscript.

1) For Q1, we have highlighted the eventual use of near-organ fine neural interfaces for human application purposes and have discussed PEDOT applicability to humans on Pages 15 and 16 in the revised manuscript.

Page 15: *Although the goal is to use the neural interface in humans, prior to clinical applications in humans, the effects of neural interface for fine nerves should be assessed in pre-clinical rodent models.*

Page 16: Additionally, researches on PEDOT in humans demonstrated its advantages over traditional metal electrodes in high-resolution electrocorticography^{64, 65}, supporting the feasibility of PEDOT for human applications.

2) For Q8, we have clarified “thick nerves” as nerves with a diameter of millimeters, and we have listed the sciatic nerves of rats as an example on Page 4 of the revised manuscript.

Page 4: ... thick nerves (diameters in the millimeter range) ... larger tissues such as rat sciatic nerves...

3) For Q9, we have listed several pre-clinical research cases for bioelectronic modulation of nerves closer to the target organs, such as ovarian, carotid sinus nerves, greater splanchnic nerves, pancreatic nerves, and splenic nerves. The studies collectively highlight the therapeutic potential of neuromodulation in treating autoimmune diseases by targeting specific nerves in mice, rats, pigs, and human models (such as splenic and pancreatic). Corresponding discussions have been added in page 3 in the revised manuscript.

Page 3: Since unselective neurostimulation often causes undesired side effects (cough, hoarseness, voice alteration, anxiety, headache, etc.)⁴, there is an urgent demand for the selective modulation of fine peripheral nerves (< 300 μm) adjacent to the target organ, such as the pancreatic and splenic nerves^{5, 6}. This approach offers a promising method for treating refractory autoimmune, cardiovascular, and metabolic diseases while reducing side effects⁷⁻⁹.

4) For Q18, we have added discussions about the common frequency ranges for

neuromodulation on Page 16 of the revised manuscript and have clarified the “1 to 10 Hz” refers to the common deformation frequency of hydrogel after implantation rather than neuromodulation frequency on Page 7 in the revised manuscript.

Page 7: ...in a dynamic mechanical environment (with strain up to 20% and frequencies from 1 to 10 Hz) ...

Page 16: The biphasic charge-balanced rectangular current pulses (10 Hz, 0.1 mA, 1.5 ms) were applied to stimulate the vagus nerve, following the methodology of previous studies^{66,67}.

5) For Q19, we have added discussions about the effects of swelling equilibrium time, equilibrium swelling rate, and gelation time on the safety of hydrogel implantation on Page 9 of the revised manuscript.

Page 9: Compared to the equilibrium swelling rate, gelation time is more crucial for the implantation feasibility of the hydrogel. In contrast, the equilibrium swelling rate and structural stability are more critical for its long-term safety. An appropriate gelation time provides operators with sufficient time to complete the injection and positioning without issues caused by rapid gelation. This ensures the hydrogel is evenly distributed at the target site and remains stable, maintaining its structure and function. Additionally, the equilibrium swelling rate and structural stability accurately reflect the hydrogel's stability after implantation. Although the ICAA hydrogel takes time to reach swelling equilibrium, its adjustable gelation time (42.4 s to 252.1 s), low equilibrium swelling rate, and structural stability ensure both procedural feasibility and post-implantation safety.

6) For Q25, we have added a discussion on the reasons for nerve tissue damage in the

positive control group on Page 16 of the revised manuscript.

Page 16: In contrast, the positive control group showed significant Schwann cell reduction due to nerve damage, primarily caused by size mismatch and suture compression. Individual nerve size variances prevent a perfect fit of standard-sized cuff electrodes, and suture fixation often exacerbates nerve compression.

7) For Q27, we have added a clearer discussion detailing whether the heart rate of rats changes during neuromodulation on Page 16.

Page 16: We observed no significant changes in the rats' weight or heart rate during the vagus nerve stimulation treatment (Supplementary Fig. 21). Additionally, typical side-effects such as coughing, hoarseness, or alterations in voice were absent. These results align with previous studies that utilized low-dose vagus nerve stimulation^{66, 67}, likely attributable to our employment of low-dose stimulation.

8) For Q28, we have added a clearer discussion about side effects during neuromodulation on Page 17.

Page 17: Additionally, typical side-effects such as coughing, hoarseness, or alterations in voice were absent. These results align with previous studies that utilized low-dose vagus nerve stimulation^{66, 67}, likely attributable to our employment of low-dose stimulation.

Reviewer #3: *This manuscript developed highly-stable, injectable, conductive hydrogel bioelectronics for chronic neuromodulation. Given the novelty of the synergistic therapeutic strategy and the comprehensive supplementary experimental*

data, I would recommend accepting this manuscript.

Response: We thank the reviewer for the constructive feedback on our manuscript.

List of Changes according to the Comments

1. The title has been changed into “Highly-stable, injectable, conductive hydrogel for chronic neuromodulation”.
2. We have listed several pre-clinical research cases for bioelectronic modulation of nerves closer to the target organs, such as ovarian, carotid sinus nerves, greater splanchnic nerves, pancreatic nerves and splenic nerves. Corresponding discussions have been added on Page 3 of the revised manuscript.
3. “Thick nerves” have been clarified as nerves with a diameter of millimeters, and we have listed the sciatic nerves of rats as an example on Page 4.
4. We have added discussions about the common frequency ranges for neuromodulation on Page 7 of the revised manuscript and have clarified the “1 to 10 Hz” referred to the common deformation frequency of hydrogel after implantation rather than neuromodulation frequency on Page 16 in the revised manuscript.
5. We have added discussions about the effects of swelling equilibrium time, equilibrium swelling rate, and gelation time on the safety of hydrogel implantation on Page 9.
6. The eventual use of near-organ fine neural interfaces for human application purposes has been highlighted on Page 15.
7. The discussion about PEDOT applicability to humans has been added on Page 16.
8. We have added a discussion on the reasons for nerve tissue damage in the positive

control group on Page 16 of the revised manuscript.

9. We have added a clearer discussion detailing whether the heart rate of rats changes during neuromodulation on Page 16.
10. We have added a clearer discussion about side effects during neuromodulation on Page 17.
11. We have made changes according to the “Author Checklist” and have prepared the required documents.